# The mechanistic and functional profile of the therapeutic anti-IgE antibody ligelizumab differs from omalizumab

Pascal Gasser [1,2,3,6], Svetlana S. Tarchevskaya[4,6], Pascal Guntern [1,2,3], Daniel Brigger[1,2], Rahel Ruppli[1,2], Noemi Zbären[1,2], Silke Kleinboelting[4], Christoph Heusser[5], Theodore S. Jardetzky [4,6]* & Alexander Eggel [1,2,6]*

Targeting of immunoglobulin E (IgE) represents an interesting approach for the treatment of allergic disorders. A high-affinity monoclonal anti-IgE antibody, ligelizumab, has recently been developed to overcome some of the limitations associated with the clinical use of the therapeutic anti-IgE antibody, omalizumab. Here, we determine the molecular binding profile and functional modes-of-action of ligelizumab. We solve the crystal structure of ligelizumab bound to IgE, and report epitope differences between ligelizumab and omalizumab that contribute to their qualitatively distinct IgE-receptor inhibition profiles. While ligelizumab shows superior inhibition of IgE binding to FcεRI, basophil activation, IgE production by B cells and passive systemic anaphylaxis in an in vivo mouse model, ligelizumab is less potent in inhibiting IgE:CD23 interactions than omalizumab. Our data thus provide a structural and mechanistic foundation for understanding the efficient suppression of FcεRI-dependent allergic reactions by ligelizumab in vitro as well as in vivo.

[1] Department of BioMedical Research, University of Bern, Bern, Switzerland. [2] Department of Rheumatology, Immunology and Allergology, University Hospital Bern, Bern, Switzerland. [3] Graduate School of Cellular and Biomedical Sciences, University of Bern, Bern, Switzerland. [4] Department of Structural Biology, Stanford University School of Medicine, Stanford, CA 94305, USA. [5] Pharmaceutical Research, Novartis AG, 4002 Basel, Switzerland. [6] These authors contributed equally: Pascal Gasser, Svetlana S. Tarchevskaya, Theodore S. Jardetzky, Alexander Eggel. *email: tjardetzky@stanford.edu; alexander.eggel@dbmr.unibe.ch

As a key driver in the development and manifestation of hypersensitivity reactions against normally non-hazardous substances, immunoglobulin E (IgE) has become a major target of therapeutic intervention strategies[1–3]. IgE is known to interact with two major receptors, FcεRI and CD23/FcεRII[4], which are involved in different immunological processes[5]. Binding of allergen-specific IgE to FcεRI expressed on immunological effector cells including basophils and mast cells occurs with high affinity ($K_D$ $10^{-10}$ M). This interaction occurs via two asymmetric binding sites on the receptor and is stabilized through the induction of a conformational change in IgE[6–8]. Exposure to allergens induces cross-linking of IgE-bound FcεRI resulting in immediate activation of allergic effector cells, which culminates in cellular degranulation and the release of vasoactive and pro-inflammatory mediators[9].

While this FcεRI-dependent cellular degranulation process accounts for immediate hypersensitivity reactions and the induction of clinical allergy symptoms, the interaction of IgE with CD23 has been reported to be involved in antigen presentation, the transport of antigens across airway and intestinal epithelial barriers and the regulation of IgE synthesis[10–13]. Binding of IgE to monomeric CD23 is of low affinity ($K_D$ $10^{-6}$–$10^{-7}$ M)[14]. However, cell surface CD23 is prone to oligomerization leading to enhanced binding of IgE:allergen complexes on antigen presenting cells[10,15]. Furthermore, various studies have provided evidence that IgE binding to CD23 on B-cells negatively regulates IgE synthesis[11,13,16].

Over the last decades, various anti-IgE inhibitors including antibodies[17–22], DARPin® proteins[23–25] and nanobodies[26,27] have been generated and tested in pre-clinical studies. To date, the monoclonal antibody omalizumab (Xolair®) represents the only licensed anti-IgE compound for clinical use. Omalizumab shows remarkable therapeutic efficacy in allergic asthma and chronic spontaneous urticaria[17,28,29]. Recently, a next-generation high-affinity anti-IgE monoclonal antibody (ligelizumab; QGE031) has been developed with the intention of overcoming some of the limitations associated with omalizumab[18,30].

Given its current status as a potential anti-IgE therapeutic and successor to omalizumab, we sought to investigate the IgE binding characteristics of ligelizumab and its modes-of-action. Here, we report the crystal structure of ligelizumab bound to IgE, revealing that it recognizes a distinct IgE epitope only partially overlapping with that of omalizumab. Ligelizumab interacts across the IgE-Fc dimer and favors the recognition of IgE in an open conformation different from its FcεRI- or CD23-bound conformations. Moreover, it binds IgE with significantly higher affinity than omalizumab and shows a correspondingly enhanced inhibition of IgE binding to FcεRI and basophil activation. In contrast and despite its higher affinity for IgE, ligelizumab is inferior to omalizumab in preventing IgE binding to CD23. Structural analysis indicates that differences in the ligelizumab epitope and spatial orientation on IgE contribute to this differential inhibition. We further observe that ligelizumab features the ability to reduce IgE production in peripheral blood mononuclear cell (PBMC) cultures, a process which may be mediated by its ability to bind IgE:CD23 complexes at the surface of B-cells. Together, our data provide a structural and mechanistic foundation for understanding why ligelizumab exerts a qualitatively and functionally distinct inhibition profile from omalizumab and is superior in suppressing FcεRI-dependent allergic reactions in vitro, in a passive systemic anaphylaxis mouse model in vivo and in clinical studies with chronic spontaneous urticaria patients[31].

## Results

**Ligelizumab binding characteristics.** A central mechanism of therapeutic anti-IgE antibodies is the neutralization of free serum IgE. The binding affinity for IgE may therefore be a major determinant of clinical efficacy. To assess binding kinetics of ligelizumab to human IgE, we performed surface plasmon resonance (SPR) measurements. Human monoclonal Sus11-IgE was displayed on the chip surface via the non-competitive anti-IgE capture antibody Le27 (ref. [32]). Different concentrations of ligelizumab IgG as well as its F(ab')$_2$ and Fab fragments were measured in consecutive cycles on surface displayed IgE (Fig. 1a and Supplementary Fig. 1a, b). As a comparison, the same measurements were performed with the therapeutic anti-IgE antibody omalizumab IgG and its F(ab')$_2$ or Fab fragments (Fig. 1b and Supplementary Fig. 1c, d). The individual association ($k_a$) and dissociation ($k_d$) constants were calculated using a 1:1 binding langmuir curve fitting model (Table 1, Fig. 1a, b and Supplementary Fig. 1a–d). With a $K_D$ of 35 pM, the interaction of ligelizumab Fab with IgE was ~88-fold stronger than the binding of omalizumab Fab. While the measurements of the association constants for ligelizumab or omalizumab Fab revealed rather small differences, ligelizumab featured an ~15-times lower dissociation rate. The difference in affinity was further confirmed by dose-dependent titration of ligelizumab or omalizumab IgG and its F(ab')$_2$ or Fab fragments on surface displayed IgE in a titration ELISA (Supplementary Fig. 1e–g).

IgE is a flexible protein, which undergoes conformational rearrangements depending on its interaction partner[33]. Various studies have demonstrated that IgE binds FcεRIα in an open Cε3 domain conformation[6], while CD23-bound IgE adopts a closed Cε3 conformational state[34] and that these receptor interactions are mutually exclusive[35]. We have previously engineered an IgE-Fc$_{3-4}$ variant (C335) that is trapped in a closed FcεRIα incompatible conformation[36], while the wild-type IgE-Fc$_{3-4}$ (C328) can adopt an open FcεRIα-binding state (Fig. 1c). To assess whether ligelizumab features binding preference for one of these conformational states, we measured its interaction with C328 or C335 IgE-Fc$_{3-4}$ by SPR. Ligelizumab recognized the wild-type C328 IgE-Fc$_{3-4}$ variant with ~60-fold higher affinity (Fig. 1d, e and Supplementary Table 1). In contrast, omalizumab showed similar binding to both conformations (Table 1 and Fig. 1g, h). These findings were further confirmed by ELISA, in which ligelizumab again showed preferential binding to wild-type C328 IgE-Fc$_{3-4}$, while omalizumab did not discriminate between the two conformational variants (Fig. 1f, i).

To obtain structural insight into the ligelizumab binding epitope on IgE, we solved the crystal structure of the C328 IgE-Fc$_{3-4}$ fragment bound to the single chain fragment variable (scFv) construct of ligelizumab to a resolution of 3.65 Å (Supplementary Table 2). The crystal structure (PDB ID: 6UQR) shows two ligelizumab scFvs binding across the IgE dimer, with each scFv forming interactions with both Cε3 domains (Fig. 1j, k). The majority of the ligelizumab interaction is mediated through VH domain interactions with one of the Cε3 domains, with a total buried surface area of ~1200 Å$^2$. This primary contact is formed by ~15 residues of the ligelizumab VH domain with heavy chain complementary determining region 1 (HCDR1) residues W31, Y32 and W33 forming key contacts at the center of the interface (Fig. 1l, m). By comparison, HCDR3 extends along the inner face of the Cε3 domain and makes more peripheral contacts with IgE. The ligelizumb VH domain contacts 21 residues of the IgE, centered around Q417, R419 and M430 (Fig. 1l, m). Ligelizumab contacts with the second Cε3 domain appear relatively minor, burying only ~170 Å$^2$ of accessible surface area and involving only 6 residues in VL with 4 residues in IgE. In this cross-dimer contact, ligelizumab VL residues interact with the IgE loop containing its conserved N-linked glycosylation site at residue 394. These contacts likely contribute little to overall ligelizumab affinity, but the cross-dimer interactions would restrict the

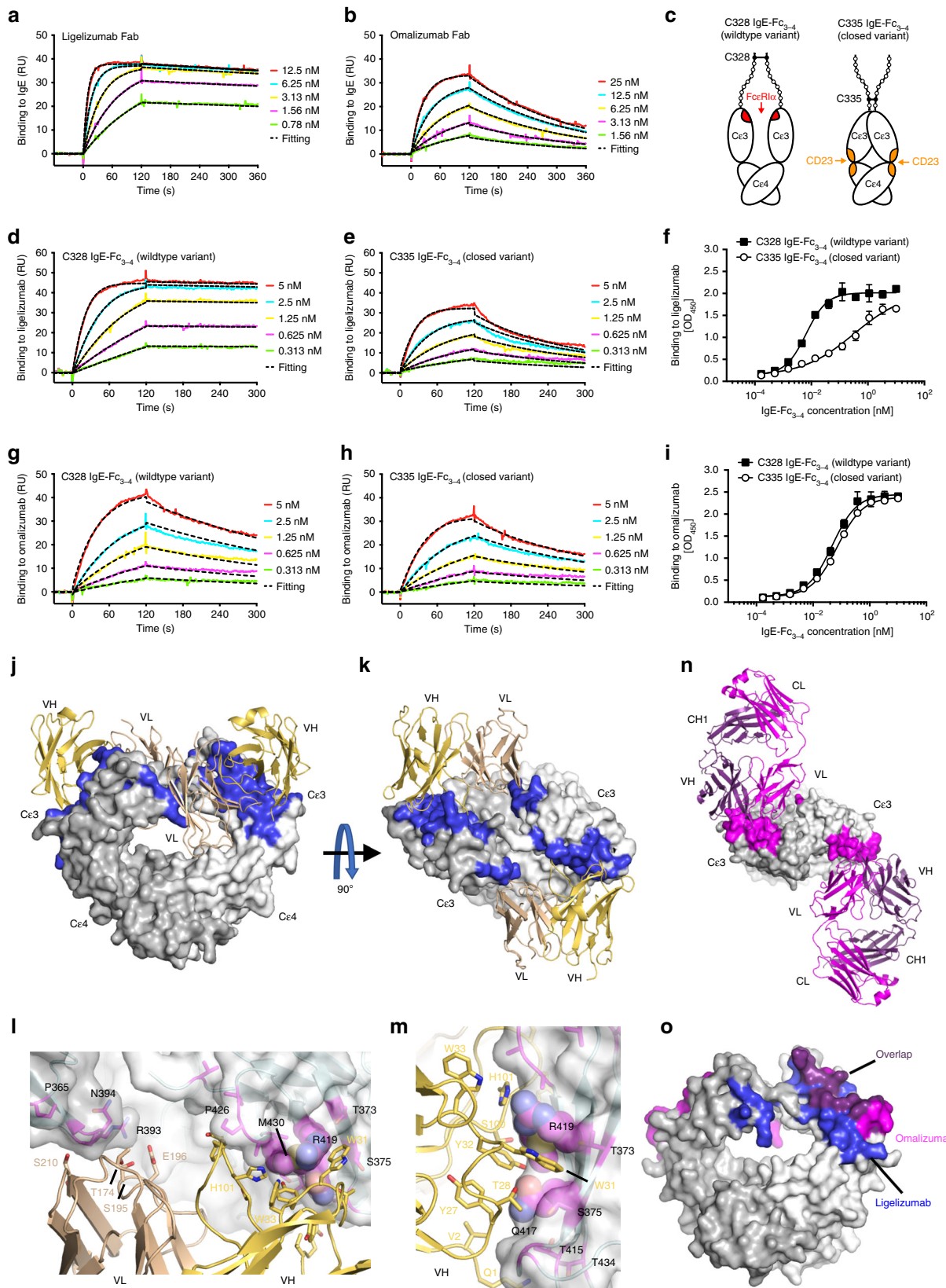

conformations accessible to the Cε3 domain. The structural data indicate that in the ligelizumab complex, the IgE Cε3 dimer appears more open, with a V336–V336 distance of 17 Å as compared to ~12 Å in the constrained C335 IgE-Fc$_{3-4}$ variant and consistent with its reduced affinity for this C335 mutant.

Ligelizumab and omalizumab clearly bind to different epitope structures on the IgE-Fc dimer; however, they share a significant overlapping region (Fig. 1n, o). In addition, the angle of binding of the two antibodies relative to the IgE-Fc domain also differs, affecting their respective functions in receptor inhibition.

**Fig. 1 Binding kinetics of ligelizumab or omalizumab on recombinant human IgE. a**, **b** Association and dissociation of ligelizumab (**a**) and omalizumab (**b**) Fab fragments to human recombinant Sus11-IgE by SPR. Each color refers to an individual measurement cycle. Curves were fitted (black dashed) using a 1:1 langmuir binding model. **c** Wildtype C328 and mutant C335 IgE-Fc$_{3-4}$. **d**, **e** Binding of wt C328 IgE-Fc$_{3-4}$ (**d**) and mut C335 IgE-Fc$_{3-4}$ variants (**e**) to ligelizumab IgG by SPR. **f** Binding of wt C328 IgE-Fc$_{3-4}$ or mut C335 IgE-Fc$_{3-4}$ to ligelizumab IgG by ELISA (technical duplicates as mean ± SEM). **g**, **h** Binding of wt C328 IgE-Fc$_{3-4}$ (**g**) and mut C335 IgE-Fc$_{3-4}$ (**h**) to omalizumab IgG by SPR. **i** Binding of wt C328 IgE-Fc$_{3-4}$ or mut C335 IgE-Fc$_{3-4}$ to omalizumab IgG by ELISA (technical duplicates as mean ± SEM). **j**–**m** Crystal structure of ligelizumab single chain Fv (scFV) bound to wt C328 IgE-Fc$_{3-4}$ (PDB ID: 6UQR). IgE-Fc$_{3-4}$ dimer in surface representation (white and light gray). Ligelizumab epitope residues shown in blue. The ligelizumab scFv is shown in cartoon format (light orange and wheat). 90° rotations with perpendicular (**j**) and parallel (**k**) view to IgE-Fc twofold axis. **l**, **m** Contact interface between ligelizumab VH and VL CDR loops with the two IgE Cε3 domains. **l** Orientation along the IgE twofold axis. **m** Orientation from the side of the VH: IgE-Fc interaction. Ligelizumab VH domain interaction is centered around IgE residues R419, M430 and Q417. Ligelizumab residue W31 packs into a pocket on IgE, with additional HCDR1 residues (Y32 and W33) forming interactions that straddle the central R419/M430/Q417 site. Interactions of the VL domain with IgE are limited and located adjacent to residue N394. **n** Structure of omalizumab bound to IgE-Fc (PDB ID: 5HYS[41]) oriented as in **k**. Omalizumab is shown in ribbon format with heavy chains (violet) and light chains (magenta). IgE is shown in surface format with interface residues (magenta). **o** Overlap of ligelizumab and omalizumab epitopes. Residues unique to omalizumab interactions (magenta) and unique to ligelizumab are (blue) and those that interact with both antibodies (violet) are shown. Source data are provided as Source Data file.

**Table 1 Binding kinetics of ligelizumab and omalizumab variants for human IgE.**

| Anti-IgE antibody | Association $k_a$ (M$^{-1}$s$^{-1}$) | Dissociation $k_d$ (s$^{-1}$) | Affinity $K_D$ (pM) |
|---|---|---|---|
| Ligelizumab IgG | $1.8 \times 10^6$ | $3.3 \times 10^{-5}$ | 17.8 |
| Omalizumab IgG | $9.1 \times 10^5$ | $2.4 \times 10^{-3}$ | 2659 |
| Ligelizumab F(ab')$_2$ | $4.2 \times 10^6$ | $5.1 \times 10^{-5}$ | 12.1 |
| Omalizumab F(ab')$_2$ | $8.7 \times 10^5$ | $2.6 \times 10^{-3}$ | 2998 |
| Ligelizumab Fab | $9.2 \times 10^6$ | $3.2 \times 10^{-4}$ | 35.0 |
| Omalizumab Fab | $1.5 \times 10^6$ | $4.6 \times 10^{-3}$ | 3090 |

**Characterization of the ligelizumab IgE inhibition profile.** Given that ligelizumab has superior affinity for IgE than omalizumab, we aimed to compare their potency in preventing IgE binding to FcεRIα using ELISA and cell-based assays. For ELISA assays, we used a Sus11-IgE concentration (0.78 nM) at the midpoint of its binding titration with FcεRIα, which falls within physiological IgE concentrations (Supplementary Fig. 2a), to investigate dose-dependent inhibition of IgE-binding to FcεRIα by ligelizumab or omalizumab IgG (Fig. 2a). The results show that ligelizumab inhibits IgE-binding to FcεRIα with a 20-fold higher potency than omalizumab. We also measured ligelizumab- and omalizumab-mediated inhibition of IgE-binding to isolated human primary basophils by flow cytometry. These cells express high levels of CD123, CD193 and carry FcεRIα-bound IgE on their surface (Fig. 2b). After removing endogenous IgE from the isolated basophils[25], we used a physiologically relevant concentration of 2 nM of JW8-IgE (Supplementary Fig. 2b) in the inhibition assay. IgE was pre-incubated with varying concentrations of ligelizumab or omalizumab IgG (Fig. 2c). Ligelizumab blocked IgE binding with greater potency, consistent with its higher affinity and with the ELISA results.

Given the difference in their binding epitopes on IgE, we further explored the ability of ligelizumab and omalizumab to inhibit IgE-binding to CD23. Due to the low affinity of the CD23 interaction, pre-formed IgE:antigen complexes were used in this ELISA. A concentration of 70 nM IgE:antigen complexes (Supplementary Fig. 2c) was incubated with physiologically relevant amounts of ligelizumab or omalizumab IgG prior to the addition to immobilized CD23. Interestingly, despite its higher affinity for IgE, ligelizumab showed a fourfold weaker inhibition of IgE-binding to CD23, as compared to omalizumab (Fig. 2d). To further evaluate CD23 competition on cells, we measured ligelizumab- and omalizumab-mediated inhibition of IgE-binding to CD23 by flow cytometry using the CD19-/CD23-expressing leukemia B-cell line RPMI8866 (Fig. 2e). A

physiologically relevant IgE concentration of 12.5 nM (Supplementary Fig. 2d) was pre-incubated with ligelizumab or omalizumab IgG in varying concentrations. In agreement with the ELISA data, we observed weaker inhibition of IgE-binding to CD23 on RPMI8866 cells with ligelizumab than with omalizumab (Fig. 2f). Thus, the two anti-IgE antibodies reveal a qualitatively distinct inhibition profile for the two IgE receptor pathways with ligelizumab being more potent and more selective for inhibition of IgE binding to FcεRI.

Previous studies have reported that human BDCA1$^+$ dendritic cells (DCs) constitutively express surface FcεRI resulting in rapid binding and endocytosis of serum IgE[37]. The mechanism of FcεRI-mediated IgE:antigen complex uptake and presentation by DCs has further been proposed to promote Th2 immune responses[38,39]. To assess the effect of ligelizumab and omalizumab on IgE-binding and IgE:antigen complex internalization with DCs, we isolated human BDCA1$^+$ DCs from whole-blood donations (Supplementary Fig. 3a) and assessed IgE binding and internalization in the presence of ligelizumab and omalizumab. In contrast to the results obtained with basophils, both anti-IgE antibodies dose-dependently inhibited binding of IgE with the same efficacy (Supplementary Fig. 3b). By analyzing IgE receptors expression on BDCA1$^+$ DCs, we found evidence for co-expression of FcεRIα and CD23, which explains the lack of superior inhibition of ligelizumab in this experimental setup (Supplementary Fig. 3c). Next, we compared the uptake of IgE:antigen complexes of basophils and BDCA1$^+$ DCs. Maximal internalization was reached after 4 h of incubation for basophils and 8 h for DCs as detected by pH-sensitive IgE staining (Supplementary Fig. 3d). In line with previous results, ligelizumab inhibited the IgE:antigen complex uptake in basophils more efficiently than omalizumab (Supplementary Fig. 3e). Interestingly, only F(ab')$_2$ fragments prevented complex internalization in DCs, whereas the full-length antibodies increased the uptake presumably through engagement of Fcγ-receptors that are present on the surface of these cells (Supplementary Fig. 3f).

The crystal structure of the ligelizumab svFv:IgE-Fc$_{3-4}$ complex provides detailed insights into its ability to inhibit FcεRI and CD23 binding (Fig. 2g–k). Views of the IgE:ligelizumab and IgE:FcεRIα complexes along the IgE dimer axis show that the two ligelizumab scFvs mimic the arrangements of FcεRI and IgE Cε2 domains on either side of the central pair of Cε3 domains (Fig. 2g, h). Therefore, at least three mechanisms could contribute to the ability of ligelizumab to efficiently block FcεRI binding. First, one of the ligelizumab scFvs occupies a largely overlapping volume with FcεRIα, indicating that substantial steric conflicts between ligelizumab and FcεRIα prevent their simultaneous binding (Fig. 2g, h). Second, the interaction footprints of ligelizumab

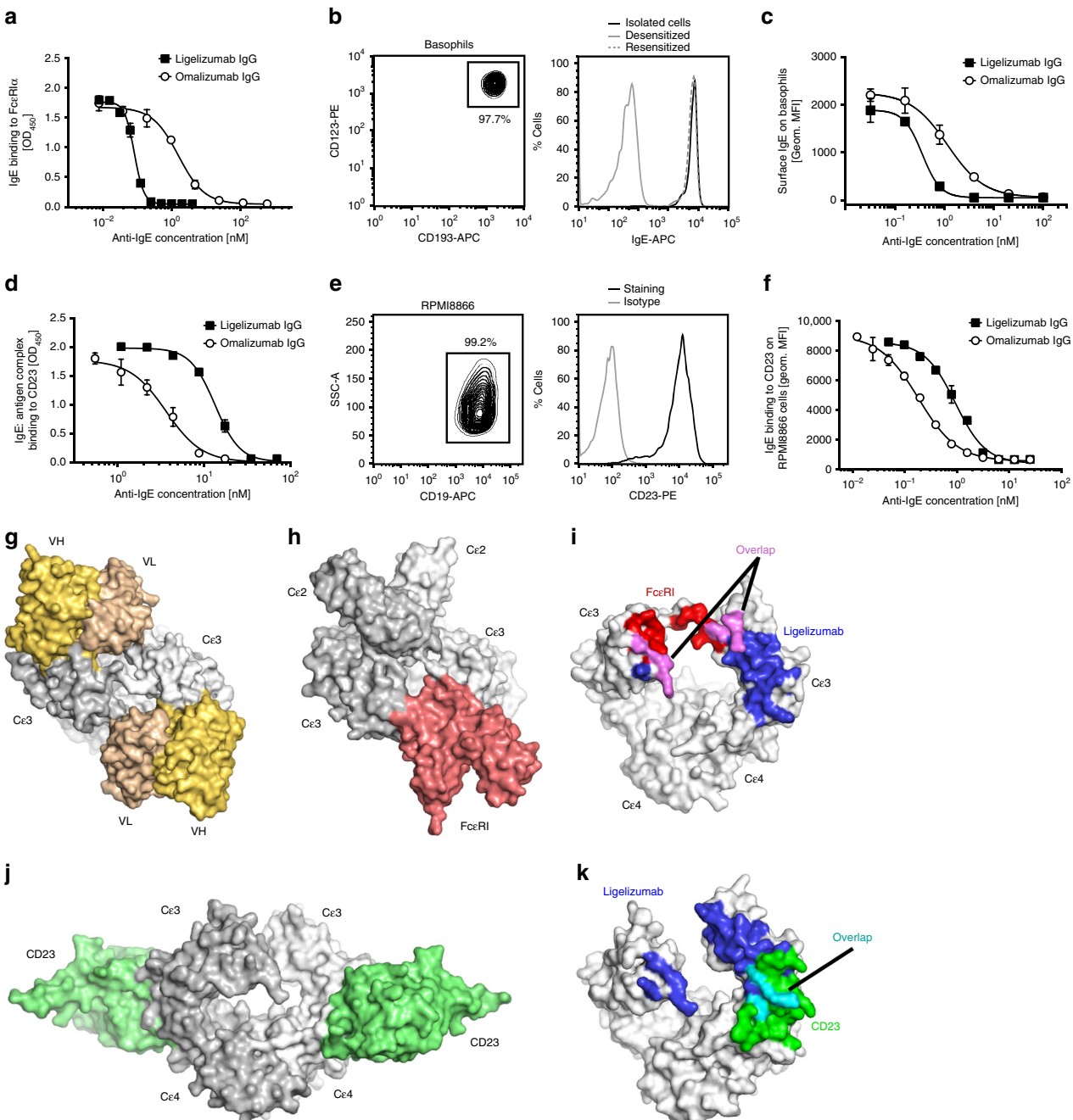

**Fig. 2 Inhibition of IgE binding to FcεRIα and CD23 by ligelizumab or omalizumab. a** Inhibition of IgE binding to FcεRIα through ligelizumab or omalizumab IgG by ELISA. **b** Isolated primary human blood basophils are gated as SSC^low, FSC^low, CD123+, CD193+ double-positive cells. A histogram of basophil surface IgE levels before (black line) and after desensitization with 5 μM disruptive-anti-IgE DARPin® bi53_79 (gray line) and after resensitization with recombinant human JW8-IgE (dashed gray line) is shown. **c** Inhibition of IgE binding to FcεRIα on isolated bi53_79-treated basophils by ligelizumab or omalizumab IgG. Binding curves were fitted using a non-linear regression model. **d** Inhibition of IgE:antigen complex binding to CD23 by ligelizumab or omalizumab IgG by ELISA. **e** Surface expression of CD23 on RPMI8866 cells by flow cytometry. **f** Inhibition of IgE binding to RPMI8866 cells by ligelizumab or omalizumab IgG. Binding curves (technical duplicates as mean ± SEM) were fitted using a non-linear regression model. **g–i** Ligelizumab sterically blocks FcεRI binding overlaps with receptor-binding residues in both IgE subunits. **g** Ligelizumab complex in surface representation with the view along the IgE twofold axis (PDB ID: 6UQR). VH and VL domains are (light orange and wheat) and IgE subunits (white and light gray) are shown. **h** The structure of IgE-Fc₂₋₄ with FcεRIα (PDB ID: 2Y7Q[53]) in surface format, with the receptor (salmon) and IgE chains (white and light gray). One ligelizumab scFv sterically blocks receptor binding, while second scFv overlaps the position of the IgE Cε2 domains. **i** Overlap of IgE residues involved in both FcεRI and ligelizumab binding. IgE-Fc₃₋₄ in a surface representation is shown with common contacts shared by ligelizumab and FcεRIα (magenta), FcεRIα-specific contacts (red) and ligelizumab-specific (blue). **j** Structure of CD23:IgE-Fc₃₋₄ complex (PDB ID: 4EZM[57]). IgE in surface representation with CD23 (light green). **k** IgE binding sites for CD23 have minimal overlap with ligelizumab epitope. IgE residues involved solely in ligelizumab binding (blue), those involved only in CD23 binding (green) and the overlapping contact residues (cyan) are shown. **a**, **c**, **d**, **f** Data shown for technical duplicates as mean ± SEM. Source data are provided as Source Data file.

and FcεRIα on IgE show that both ligands share a number of identical binding residues on IgE (Fig. 2i), indicating potential for direct, though limited, competition for subsite binding. Third, the binding of ligelizumab across the IgE dimer restricts the arrangement of the Cε3 domains into a conformation that is incompatible with FcεRI binding (Fig. 2g, h). Furthermore, ligelizumab binding to intact IgE would also displace Cε2 domains, generating a more linear structure that might interfere with FcεRIα binding[40].

CD23 interacts with IgE-Fc at the hinge region between Cε3 and Cε4 domains (Fig. 2j), favoring a closed conformation of Cε3 domains[34]. Comparison of the CD23 and ligelizumab interaction sites on IgE indicates relatively minor overlap between binding sites of these two IgE ligands (Fig. 2k). Furthermore, the orientation of the ligelizumab VHVL domains on IgE (Fig. 2g) indicates that the ligelizumab Fab would project away from the IgE and would not sterically overlap bound CD23, suggesting that competition for IgE surface subsites and the stabilization of an open Cε3 conformation would be the primary mechanism of CD23 inhibition. In contrast, omalizumab-mediated inhibition of CD23 binding to IgE is effected by both substantial steric overlap between omalizumab and CD23, and through direct competition for IgE-binding residues by the omalizumab heavy chain[41]. Comparisons of the omalizumab and ligelizumab complexes with IgE (Fig. 1j–o and Supplementary Fig. 4) show that the antibody binding footprints and orientations contribute to the significant differences in their relative abilities to inhibit CD23 or FcεRIα binding.

**Determination of ability to disrupt FcεRI:IgE complexes**. We previously reported that omalizumab in addition to its ability to neutralize free serum IgE accelerates IgE dissociation from the surface of FcεRIα-expressing cells at concentrations well above its $K_D$[24,25]. To test whether ligelizumab shares this additional mode of action, we assessed its ability to actively remove FcεRIα-bound IgE. Using SPR, we pre-complexed monoclonal human Sus11-IgE with immobilized recombinant human FcεRIα. Subsequently, 0.25–1 µM ligelizumab IgGs were continuously added for 8 h to FcεRIα-bound IgE. No dissociation above buffer baseline was observed for any of the ligelizumab concentrations (Fig. 3a). In contrast, omalizumab IgG showed dose-dependent removal of IgE from FcεRIα at concentrations ≤1 µM as judged by the steadily declining surface signal (Fig. 3b). Additionally, we evaluated ligelizumab- and omalizumab-mediated IgE dissociation from FcεRIα on the cell surface of isolated primary human basophils. Using the same concentrations of anti-IgE antibodies and antibody fragments, we quantified IgE cell surface levels after 3 and 6 days of cell culture in the presence of the respective anti-IgE antibody (Fig. 3c). Again, omalizumab but not ligelizumab treatment resulted in dose-dependent removal of surface IgE.

We have previously observed that omalizumab can form stable ternary complexes with FcεRIα-bound IgE-Fc$_{3-4}$ fragments without removing them from the receptor[25,41]. This is due to the exposure of one of the omalizumab epitopes that is buried by Cε2 domains in the intact IgE. We therefore assessed whether ligelizumab exhibits similar binding behavior using SPR. IgE-Fc$_{3-4}$ was pre-complexed with immobilized FcεRIα and ligelizumab IgG was subsequently added. Interestingly, we observed rapid disruption of IgE-Fc$_{3-4}$:FcεRIα complexes (Fig. 3d). This was not the case for omalizumab IgG, which showed pronounced binding to IgE-Fc$_{3-4}$:FcεRIα complexes without obvious disruptive activity (Fig. 3e). The anti-IgE antibody Le27[32], which binds non-competitively to a Cε4 domain epitope and was used as a control, also recognized FcεRIα-bound IgE-Fc$_{3-4}$ in a dose-dependent manner (Fig. 3f).

The structure of the IgE-Fc$_{3-4}$:ligelizumab scFv complex suggests a conformational mechanism to explain the ability of ligelizumab to disrupt these preformed IgE-Fc$_{3-4}$:FcεRIα complexes. Superposition of the ligelizumab and FcεRI complex structures through the Cε3 domain that forms the majority of the exposed ligelizumab epitope shows significantly different arrangements of the second Cε3 domain (Fig. 3g, h). While FcεRI binding requires an asymmetric arrangement of the two Cε3 domains, ligelizumab binding restricts the position of the second Cε3 domain, causing an overall shift in FcεRI-binding loops of ~11 Å (Fig. 3g, h). Ligelizumab binding forces the Cε3 domains into a more symmetrical arrangement that closely aligns with the IgE dimer twofold axis defined by the Cε4 domains and that is incompatible with FcεRI binding. The ability of ligelizumab to bind and dissociate the IgE-Fc$_{3-4}$:FcεRI complexes suggests that the complex can dynamically access conformational states in which the secondary Cε3 domain does not sterically block ligelizumab binding.

To further investigate whether ligelizumab accelerates dissociation of FcεRI-bound IgE-Fc$_{3-4}$ on allergic effector cells, we isolated primary human basophils, removed endogenous IgE from the cell surface using a disruptive anti-IgE DARPin® protein, re-sensitized the cells with either 100 nM JW8-IgE or C328 IgE-Fc$_{3-4}$ and subsequently added ligelizumab or omalizumab IgG. As expected, the IgE surface levels of JW8-IgE re-sensitized cells did not show any decrease upon treatment with either of the two anti-IgE antibodies at these concentrations as measured by flow cytometry (Fig. 3i). Additionally, we analyzed the activation status of these cells by measuring CD63 surface levels. In line with our SPR data suggesting the inability of ligelizumab or omalizumab to recognize FcεRI-bound full length IgE (Supplementary Fig. 5a–e), no activation was observed for either of the two anti-IgE antibodies (Fig. 3j). Re-sensitizing cells with IgE-Fc$_{3-4}$, instead of intact IgE, revealed that ligelizumab but not omalizumab treatment resulted in a dose-dependent reduction of surface IgE-Fc$_{3-4}$ levels on cells (Fig. 3k). Strikingly and in line with the corresponding binding data, we found that omalizumab but not ligelizumab can activate basophils re-sensitized with IgE-Fc$_{3-4}$ in a dose-dependent manner (Fig. 3l).

**Engagement of CD23:IgE complexes**. CD23 is known to play an important role in enhancing IgE-mediated allergen presentation by antigen presenting cells and in the regulation of IgE production by B-cells[5]. Various studies have demonstrated that compounds targeting CD23 or CD23-bound IgE on B-cells can inhibit IgE production[22,42–44]. Since the crystal structure of ligelizumab with IgE-Fc$_{3-4}$ showed only a minor overlap with CD23-binding residues, we assessed whether ligelizumab might also be able to bind IgE:CD23 complexes. For this purpose, we performed SPR experiments in which JW8-IgE was pre-complexed with immobilized CD23 on the chip surface (Fig. 4a). Upon subsequent injection of different ligelizumab or omalizumab concentrations, the IgE binding signal immediately decreased, indicating that IgE is displaced from CD23 by both anti-IgE antibodies (Fig. 4b). To check whether ligelizumab or omalizumab IgG formed ternary complexes with the remaining CD23-bound IgE on the chip surface, we additionally injected a polyclonal anti-IgG antibody. While CD23:IgE:ligelizumab complexation could be revealed by anti-IgG antibody, only minor signal was detectable with omalizumab (Fig. 4c). To confirm these results on a cellular level, CD23-expressing RPMI8866 cells were pre-sensitized with 12.5 nM JW8-IgE, washed and subsequently treated with an equimolar concentration of ligelizumab, omalizumab or a control IgG. Both anti-IgE treatments resulted in a reduction of surface IgE levels as measured by quantifying remaining CD23-bound IgE (Fig. 4d).

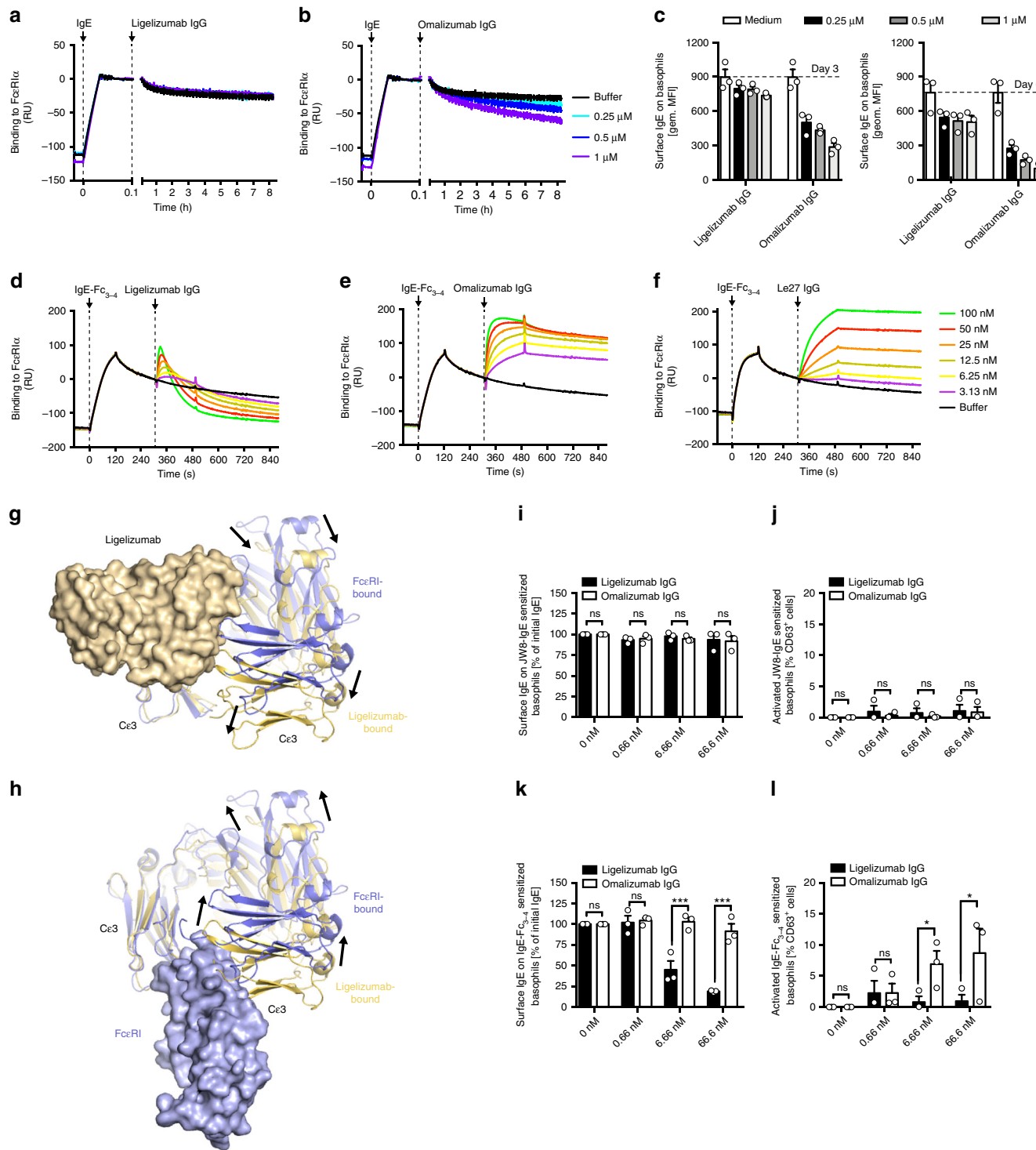

In line with the increased affinity of ligelizumab for free IgE, it showed more pronounced reduction of CD23-bound IgE cell surface levels. Nevertheless, we detected more prominent binding of ligelizumab to IgE remaining on the cell surface (Fig. 4e), which is in line with the SPR results. Additionally, when CD23 was blocked with an anti-CD23 compound prior to sensitization of the cells with JW8-IgE, almost no surface IgE could be detected and only minor binding of ligelizumab was observed, providing further evidence that ligelizumab binding to the RPMI8866 surface is dependent on the presence of CD23:IgE complexes. By adding 12.5 nM JW8-IgE and an equimolar amount of ligelizumab to RPMI8866 cells, roughly one-third of

the cells showed ligelizumab-binding on their surface (Fig. 4f, g). We additionally performed image stream flow cytometry analysis with the same experimental setup to provide direct visual evidence for ligelizumab-binding on IgE-sensitized RPMI8866 cells. All conditions showed clear staining for CD23 and IgE. Ligelizumab-treated cells were additionally positive for IgG staining (Fig. 4h) and showed clustering of IgE, indicating co-aggregation of CD23-bound IgE by ligelizumab (Fig. 4i). Co-localization analysis of IgE and IgG staining in ligelizumab-treated cells resulted in a 73.7% overlapping signal, while Omalizumab again showed only very weak binding and negligible co-localization.

**Fig. 3 Disruption of FcεRI:IgE complexes by ligelizumab or omalizumab. a** and **b** Removal of FcεRI pre-complexed human recombinant Sus11-IgE by ligelizumab (**a**) or omalizumab (**b**) IgG by SPR. Each color refers to an individual measurement cycle. Black line refers to baseline buffer control. Black arrows indicate time of injection. **c** Isolated primary human basophils ($n = 3$ donors) were incubated for 3 or 6 days with indicated concentrations of ligelizumab and omalizumab IgG. Levels of cell surface IgE were quantified by flow cytometry (shown as mean ± SEM). **d–f** Interaction of ligelizumab or omalizumab IgG with FcεRI-bound IgE-Fc$_{3-4}$. **d** and **e** Binding of ligelizumab (**d**) and omalizumab (**e**) IgG to FcεRI pre-complexed human recombinant wt C328 IgE-Fc$_{3-4}$ by SPR. **f** The monoclonal anti-IgE antibody Le27 was included as positive control. Each color refers to an individual measurement. Black line refers to baseline buffer control. Black arrows indicate the time of injection. **g**, **h** Comparisons of ligelizumab- and FcεRIα-IgE complex structures. Complexes were superimposed using the primary VH-interacting Cε3 domain (corresponding to FcεRIα site 2), predominantly exposed in the IgE-Fc$_{3-4}$ receptor-bound complex because of the absent Cε2 domains. A single ligelizumab (**g**) and FcεRIα (**h**) are shown in surface representations (light orange and light blue). Two conformations of the IgE-Fc are shown. **g** Binding of ligelizumab restricts secondary Cε3 domain conformation, preferring displacement (downward arrows). **h** FcεRIα binding to ligelizumab-stabilized Fc conformation is sterically blocked, requiring displacement of the Cε3 domain (upward arrows). Surface IgE levels (**i**) and percentage of activated cells (**j**) of JW8-IgE sensitized primary human basophils incubated for 30 min with ligelizumab or omalizumab IgG. Surface IgE levels (**k**) and percentage of activated cells (**l**) of IgE-Fc$_{3-4}$ sensitized primary human basophils incubated for 30 min with ligelizumab or omalizumab IgG. **i–l** Biological triplicates are displayed as mean ± SEM. Different conditions were compared to each other using two-way ANOVA with Sidak's multiple comparison. *$P < 0.05$, ***$P < 0.001$, ns = not significant. Source data are provided as Source Data file.

**Assessment of functional efficacy.** Given the differences in binding affinities, epitopes and receptor competition profiles of ligelizumab and omalizumab, the two anti-IgE antibodies were further compared in functional assays. First, we assessed their inhibitory efficacy in a basophil activation test. To do so, endogenous IgE was removed from isolated primary human basophils using a disruptive anti-IgE DARPin® protein and cells were reloaded with different concentrations of NIP-specific JW8-IgE. Upon challenge of the resensitized cells with NIP$_7$-BSA antigen, basophil degranulation was quantified by measuring cell surface CD63 levels using flow cytometry (Fig. 5a). IgE dose-dependent basophil activation was observed at a constant antigen concentration of 100 ng/ml NIP$_7$-BSA (Supplementary Fig. 2e). A concentration of 0.68 nM JW8-IgE was subsequently used for pre-incubation of ligelizumab or omalizumab IgG with JW8-IgE prior to re-sensitization of the cells. Antigen stimulation with NIP$_7$-BSA showed dose-dependent inhibition of basophil activation with both anti-IgE antibodies, whereby ligelizumab was more potent than omalizumab (Fig. 5b), consistent with the greater inhibitory activity of ligelizumab in blocking the IgE: FcεRIα interaction.

To further substantiate these findings in a representative FcεRI-dependent in vivo allergy model, a passive systemic anaphylaxis test was performed using mice transgenic for the human FcεRIα (huFcεRIα tg)[45]. These mice can be passively sensitized with human antigen-specific IgE and challenged with the corresponding antigen to induce a systemic anaphylaxis (PSA). The mice were pre-treated with 10 μg of anti-IgE antibody or PBS prior to sensitization with 20 μg of NIP-specific JW8-IgE (Fig. 5c). The following day the mice were challenged with 200 μg of NIP$_{20}$-BSA and the body core temperature was measured. While mice treated with ligelizumab were completely protected against antigen-induced systemic anaphylaxis, omalizumab-treated animals showed only partial protection (Fig. 5d). These data reflect the degree of remaining IgE found on mast cells in peritoneal lavages from the mice, as identified by CD45$^+$, c-kit$^+$, CD200R3$^+$ cells by flow cytometry (Fig. 5e). Thus, the cell surface IgE levels closely correlate with the functional results of the PSA (Fig. 5f), with the ligelizumab-treated mice displaying only low levels of IgE on their peritoneal mast cells. These data further demonstrate the improved activity of ligelizumab in neutralizing free IgE and inhibiting FcεRI-dependent allergic reactions in vivo.

A recent study has provided strong evidence that ligelizumab suppresses free serum IgE levels in humans for a significantly longer period-of-time than omalizumab after single dose injection[18]. To assess whether this effect might additionally be related to ligelizumab-mediated inhibition of IgE production by B-cells, we performed IgE ELISpot assays. In vitro IgE synthesis is induced through incubation of isolated human PBMCs with IL-4 and anti-CD40 antibody[46]. Following this stimulation, we determined the number of IgE-producing B-cells in the cell culture by ELISpot assays and determined soluble IgE in culture supernatants. Incubation of the PBMCs with ligelizumab resulted in a significant reduction of IgE-producing B-cells in PBMC cultures, whereas the inhibition with omalizumab was less pronounced (Fig. 5g, h). In line with this finding, soluble IgE in culture supernatants of ligelizumab IgG or F(ab')$_2$-treated PBMCs was significantly decreased. Again, omalizumab IgG showed less efficient suppression and omalizumab F(ab')$_2$ had no effect on IgE production, strongly suggesting that its suppression is mediated via Fcγ-receptors. Interestingly, blocking of CD32b (FcγRIIb) in PBMC cultures completely abrogated omalizumab-mediated suppression of IgE production, whereas ligelizumab-mediated suppression remained unaltered. Together our results demonstrate that the distinct IgE binding profiles and inhibition properties of ligelizumab and omalizumab significantly impact their functional activity.

## Discussion

Here, we compare structural and functional studies of two anti-IgE antibodies, ligelizumab and omalizumab, to better understand how their interactions with IgE impacts inhibitory mechanisms and predicts their potential therapeutic benefit. Ligelizumab and omalizumab recognize distinct binding epitopes in the IgE Cε3 domain with some overlap and show different sensitivities to IgE conformation. Therefore, the two anti-IgE antibodies display different abilities to inhibit IgE interactions with FcεRI and CD23 and feature a qualitatively distinct inhibition profile. Consequently, they greatly differ in their functional activities in blocking effector cell activation and IgE synthesis. While the increased affinity of ligelizumab for IgE explains superiority over omalizumab regarding neutralization of free serum IgE, we have identified an additional mode of action for ligelizumab through the inhibition of IgE production, which may provide additional therapeutic benefit. We observe that ligelizumab is more efficient in suppressing FcεRI-dependent allergic reactions in an in vivo model, while omalizumab may have advantages in blocking antigen presentation and transport processes that are dependent on IgE:CD23 interactions[47,48]. This balance of modulating IgE interactions with its two receptors, along with their associated functions, might have future consequences for the design of anti-IgE therapeutics.

Our data confirm that ligelizumab does not recognize FcεRIα-bound IgE, which represents the critical safety requirement for therapeutic anti-IgE antibodies. Interestingly, we observed that ligelizumab efficiently removes IgE-Fc$_{3-4}$ fragments from FcεRIα,

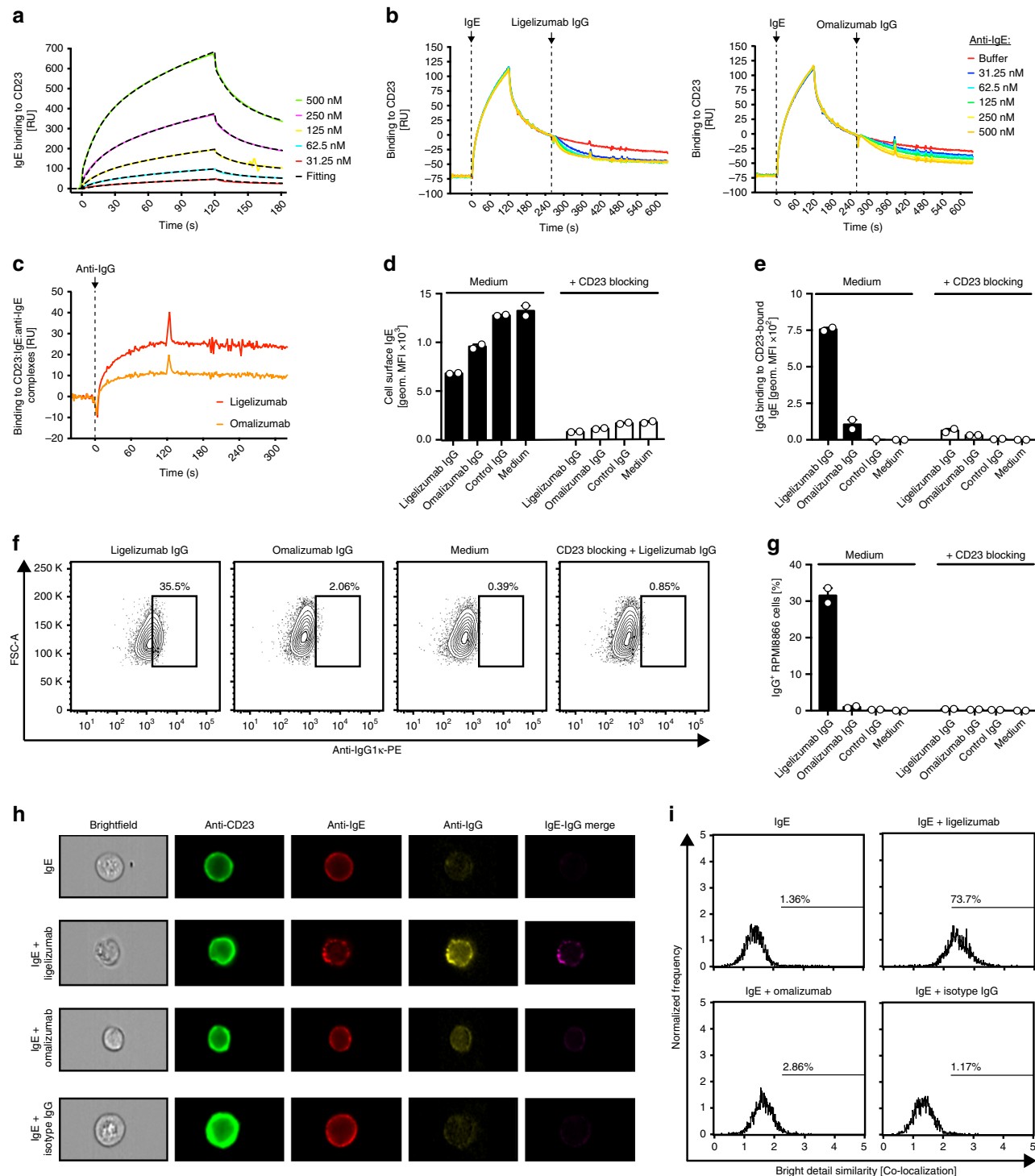

**Fig. 4 Binding of CD23:IgE complexes by ligelizumab or omalizumab. a** Association and dissociation of IgE to immobilized human recombinant CD23 by SPR. Each color refers to an individual measurement cycle. Curves were fitted (black dashed lines) using a two-state reaction binding model. **b** Binding of ligelizumab and omalizumab IgG to CD23 pre-complexed human recombinant IgE was measured by SPR. **c** Binding of polyclonal anti-IgG to CD23:IgE:anti-IgE complexes was assessed by SPR. **d** Flow cytometric quantification of IgE (**d**) and IgG (**e**) surface levels on CD23-expressing RPMI8866 cells after treatment with ligelizumab, omalizumab or isotype IgG with or without blocking of CD23 prior to IgE sensitization (data for technical duplicates are displayed as mean ± SEM). Representative flow cytometry plots (**f**) and quantification of IgG-positive RPMI8866 cell frequencies (**g**) after treatment with ligelizumab or omalizumab with or without blocking of CD23 prior to IgE sensitization (data for technical duplicates are displayed as mean ± SEM). Representative image stream flow cytometry pictures for CD23-, IgE- and IgG staining (**h**) and co-localization analysis for IgE- and IgG staining (**i**) of IgE-sensitized RPMI8866 cells after treatment with ligelizumab, omalizumab or isotype IgG. Source data are provided as Source Data file.

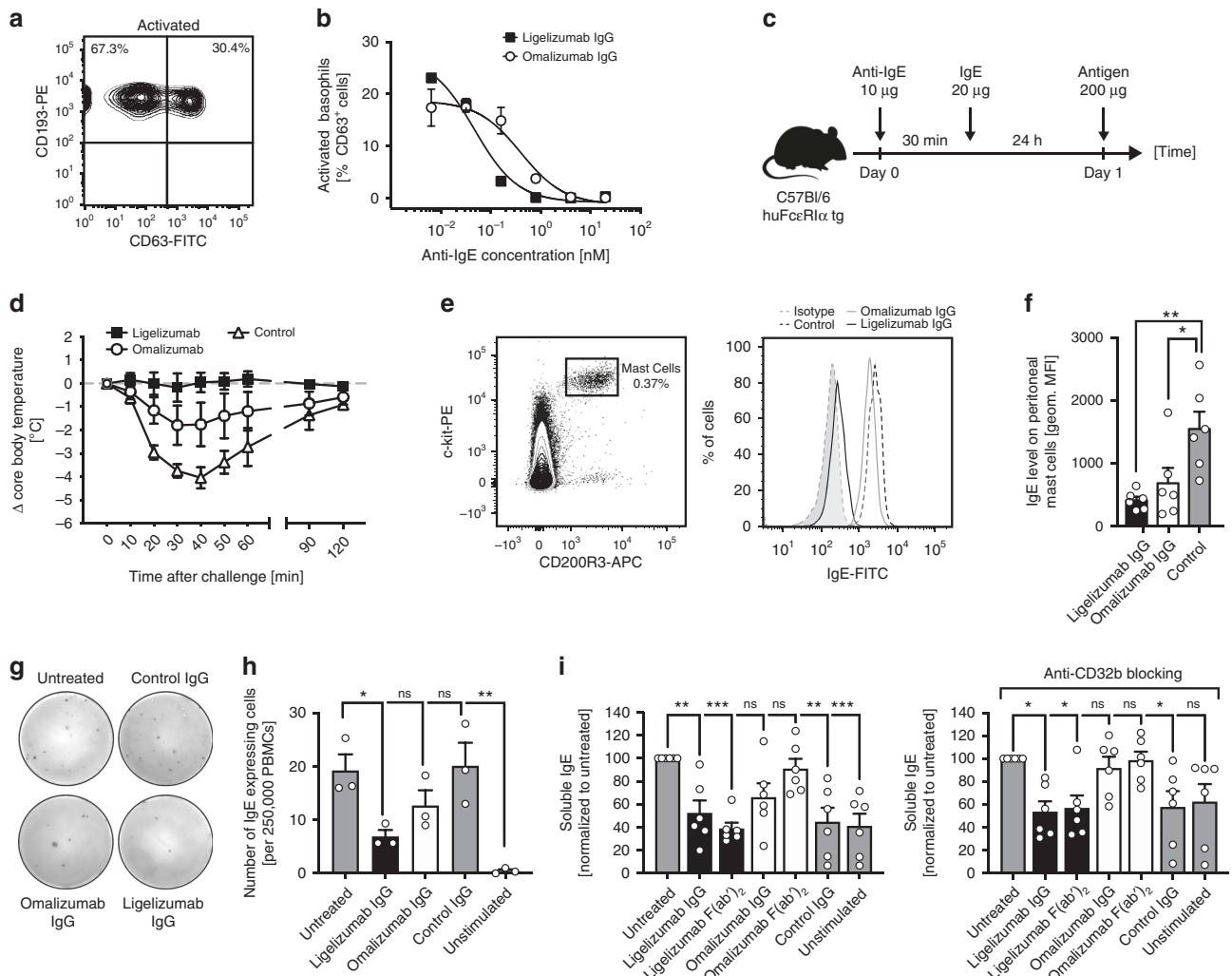

**Fig. 5 Inhibition of basophil activation and IgE production of B-cells by ligelizumab or omalizumab. a** NIP$_7$-BSA-mediated activation of isolated CD193$^+$ primary human basophils resensitized with JW8-IgE using CD63 as degranulation marker by flow cytometry. **b** Dose-dependent inhibition of primary human basophil activation by pre-incubation of JW8-IgE with increasing concentrations of ligelizumab or omalizumab IgG and subsequent stimulation of the cells with 100 ng/ml NIP$_7$-BSA. Activation curves for technical duplicates were fitted using a non-linear regression. **c** Experimental setup for the passive systemic anaphylaxis assay with huFcεRIα tg mice. **d** Changes in body core temperature after antigen challenge for mice treated with ligelizumab, omalizumab or PBS as control ($n = 6$ per group; pooled data from two individual experiments). Representative flow plots (**e**) and absolute quantification (**f**) of flow cytometric analysis of peritoneal mast cells after passive systemic anaphylaxis. **g** Representative pictures of IgE ELISpots with PBMCs from human donors ($n = 3$ donors) that were stimulated with human recombinant IL-4 and anti-human CD40 antibody in the presence of a control IgG antibody, ligelizumab or omalizumab IgG. **h** Quantification of the number of IgE expressing cells. **i** Quantification of soluble IgE in cell culture supernatants of PBMCs from human subjects ($n = 6$ donors) that were stimulated with human recombinant IL-4 and anti-human CD40 antibody in the presence of ligelizumab IgG or F(ab')$_2$ fragments, omalizumab IgG or F(ab')$_2$ fragments or a control IgG antibody. Data are shown as mean ± SEM. Different treatments were compared relative to the untreated group using one-way ANOVA with Dunnett's multiple comparison referenced to untreated controls. *$P < 0.05$, ***$P < 0.001$. Source data are provided as Source Data file.

while omalizumab binds such complexes without disrupting them[25,41]. Furthermore, we could show that the interaction of omalizumab with such FcεRI-bound IgE-Fc$_{3-4}$ fragments on primary human basophils results in a dose-dependent degranulation of the cells. Whether such IgE fragments exist in vivo in certain situations remains to be investigated as well as whether this could potentially explain rare systemic anaphylactic reactions in roughly 0.2% of omalizumab-treated patients[49]. Ligelizumab lacks the ability to cross-link FcεRIα-bound IgE-Fc$_{3-4}$ fragments and it will be interesting to see whether it shows a reduced risk of anaphylaxis and an altered safety profile as compared to omalizumab.

We further observe that omalizumab inhibits IgE-binding to CD23 more potently than ligelizumab, which could potentially

play an important role in the inhibition of IgE-mediated antigen presentation and IgE-mediated transport across epithelial barriers[12,50]. It has been shown that allergen-IgE complex-mediated eosinophilic lung inflammation in mice was CD23 dependent[48]. These findings might explain why ligelizumab treatment lacked superior efficacy in phase 2 clinical trials with severe asthma patients compared to placebo and omalizumab (NCT02075008). On the other hand, ligelizumab has the ability to recognize CD23-bound IgE on B-cells and decreases IgE production in PBMC cultures. This effect was not mediated by ADCC as F(ab')2 fragments revealed a similar inhibition of IgE production. Our data indicating that ligelizumab downregulates IgE production is in line with the finding that single dose injection of ligelizumab resulted in a significantly longer suppression of free serum IgE

compared to omalizumab[18]. Further, our results strongly suggest that the mechanism of ligelizumab suppression of IgE synthesis is independent of Fcγ-receptor engagement but might rather be due to its ability to bind and aggregate CD23:IgE complexes on the surface of B-cells, which has not been observed for omalizumab.

In summary, the structural and mechanistic differences that we have found between ligelizumab and omalizumab may have different impacts on FcεRI- and CD23-mediated pathways in patients. Presently, it is concluded that ligelizumab has the potential to be particularly efficacious in diseases driven predominantly by FcεRI-dependent reactions, as observed in CSU[31].

## Methods

**Recombinant proteins and antibodies.** Ligelizumab as well as omalizumab antibodies and fragments were produced and kindly provided by Novartis Pharma AG (Basel, Switzerland). Sus11-IgE, JW8-IgE and the anaphylactogenic monoclonal anti-IgE antibody Le27 were purchased from NBS-C BioScience (Vienna, Austria). Monoclonal anti-human CD40 antibody was purchased (Enzo Life Sciences, NY, USA). Recombinant extracellular part of human FcεRIα as well as the wild-type C328 IgE-Fc$_{3-4}$ and the mutated C335 IgE-Fc$_{3-4}$ were produced in our laboratory[36]. Recombinant human CD23 was purchased (R&D Systems, Minneapolis, MN, USA). Recombinant human IL-3 and IL-4 were purchased from Peprotec (London, UK). Cells were cultured in RPMI$^{+/+}$ medium composed of RPMI 1640 medium (Biochrome, Cambridge, UK) complemented with 10% Hyclone FCS (Fisher Scientific, NH, USA), penicillin 100 U/ml, 100 μg/ml streptomycin (100× penicillin/streptomycin, Merck, Darmstadt, Germany) and 10 mM HEPES buffer (stock-solution 1 M, Life Technologies, CA, USA). For flow cytometry, we used the following antibodies: anti-human IgE FITC (clone Ige21, Thermo Fisher Scientific, MA, USA), monoclonal mouse anti-human FcεRIα APC (clone AER-37, Thermo Fisher Scientific, MA, USA) and the appropriate isotype controls monoclonal mouse IgG1, κ Isotype control FITC (Thermo Fisher Scientific, MA, USA) and mouse IgG2b Isotype control APC (Thermo Fisher Scientific, MA, USA), monoclonal rat anti-mouse CD200R FITC (clone OX-110, Bio-Rad, CA, USA), monoclonal rat anti-mouse CD117 PE (clone 2B8, Thermo Fisher Scientific, MA, USA), monoclonal mouse anti-human CD19 APC (clone HIB19, BD Bioscience, USA), monoclonal mouse anti-human CD23 FITC and PE (clone EBVCS-5, Biolegend, CA, USA), monoclonal mouse anti-human Ig kappa light chain PE (clone TB28-2, eBioscience, CA, USA). For basophil activation testing, the anti-human CCR3 and anti-human CD63 antibody staining mix from the Flow CAST® kit was used (Bühlmann Laboratories AG, Schönenbuch, CH).

**Expression and purification of IgE:ligelizumab complexes.** A scFv codon-optimized construct of ligelizumab was synthesized (Genscript) and cloned into EcoRI/BamHI sites in the pTTVH8G vector for expression in mammalian HEK 293-6E cells (National Research Council, NRC, Canada). The scFv construct consists of the VEGF signal sequence, the ligelizumab VH domain, a GTG (GSGGG)$_3$AS linker, the ligelizumab VL domain, a TEV cleavage site and linker to a His$_8$ tag (MNFLLSWVHWSLA LLLYLHHAKWSQAAPMAEGGGQNQVQLV QSGAEVMKPGSSVKVSCKASGYTFSWYWLEWVRQAPGHGLEWMGEIDPGT FTTNYNEKFKARVTFTADTSTSTAYMELSSLRSEDTAVYYCARFSHFSGSNYD YFDYWGQGTLVTVSSGTGGSGGGGSGGGGSGGGASEIVMTQSPATLSVSPG ERATLSCRASQSIGTNIHWYQQKPGQAPRLLIYYASESISGIPARFSGSGSGTEF TLTISSLQSEDFAVYYCQQSWSWPTTFGGGTKVEIKENLYFQSGGSGHHHHH HHH). The wild-type IgE-Fc$_{3-4}$ with a VGEF signal sequence derived from the pTTVH8G vector was cloned into pYD7 vector (NRC, Canada).

The ligelizumab scFv was co-expressed with the human IgE-Fc$_{3-4}$ by co-transfection in mammalian HEK 293-6E cells. A 1:1 mixture of plasmids was transfected using linear polyethylenimine (PEI), 25 KD (Polysciences) in a DNA: PEI ratio of 1:3. The proteins were expressed for 100–120 h. After harvesting, the cell supernatant was filtered through a 0.45-μm filter (Millipore) and incubated overnight at 4 °C with Ni-NTA resin (Qiagen), washed with wash buffer (50 mM Tris pH 8.0, 300 mM NaCl, 5 mM imidazole) and eluted with elution buffer (50 mM Tris pH 8.0, 300 mM NaCl, 200 mM imidazole). Eluted protein complex was concentrated using an Amicon Ultra-15 (Millipore) and purified with gel filtration on a Superdex 200 10/300 GL column (GE) using a running buffer consisting of 20 mM Tris pH 8.0, 150 mM NaCl. His-tags were removed by cleavage with TEV protease in the presence of 2.5 mM BME.

**Crystallization and structure determination.** Purified scFv:IgE-Fc$_{3-4}$ complex was concentrated to a final concentration of 7.3 mg/ml in 20 mM Tris pH 8.0. Crystallization was carried out using the hanging drop method, with a precipitant composed of 0.2 M Na thiocyanate pH 6.9, 20% PEG3350 and 10 mM spermidine as an additive. Crystals were obtained in 2–5 days at 14 °C. Crystals were harvested and frozen in 0.2 M Na thiocyanate, 23% PEG 3350, 25% glycerol.

A complete diffraction dataset for the complex crystal was collected at 100° K at the Advanced Photon Source at Argonne National Laboratories LS-CAT beamline using an X-ray wavelength of 0.979 Å. The diffraction data were processed with X-

Ray Detector Software (XDS)[51]. The crystals belong to space group P2$_1$2$_1$2$_1$ (Supplementary Table 2). The resolution limit of the data (3.65 Å) was set using an I/sigma value of 1.3 Å and a CC1/2 value of 0.54. Molecular replacement was carried out with Phenix Phaser[52] using the structure of the IgE-Fc$_{2-4}$ (Protein Data Bank code: 2Y7Q[53]) as a search model for IgE and the scFv structure of the anti-CD277 antibody 103.2 (Protein Data Bank code: 4F9P[54]) as a search model for ligelizumab. Manual model building was carried out in Coot[55] and refinement was done in Phenix Refine[52] using data from 3.65 to 20.0 Å. Model quality was analyzed by Coot, and structure figures were generated using PyMol (Schrodinger, LLC, New York). The final model has Ramachandran statistics of 97.3% of residues in the preferred region, 2.7% of the residues in the additionally allowed region and no residues in the disallowed region. Additional refinement statistics are collected in Supplementary Table 2.

**Protein interaction measurements with SPR.** All SPR measurements were carried out on a GE Healthcare Biacore X100 device (IL, USA). HBS-EP$^+$ was used as running buffer at a flowrate of 10 μl/min. The target proteins were immobilized on flow cell 2 (Fc2) of a CM5 sensor chip by standard amine coupling. The sensorgrams reflect binding responses on Fc2 minus binding responses on the reference Fc1. To determine binding kinetics, we used the BIAevaluation software. Affinity constants were calculated using a 1:1 langmuir curve fitting model.

To determine binding kinetics of ligelizumab and omalizumab antibodies/fragments for human full-length IgE, 3000 RU of the anti-IgE Le27 were immobilized on flow cell 2 at pH 4.0. A concentration of 30 nM Sus11-IgE was subsequently captured for 120 s to reach a response of ~100 RU. Various concentrations (1.56–25 nM) of ligelizumab and omalizumab antibodies/fragments were injected for 120 s and the dissociation was measured for 240 s under constant buffer flow. After each run, the chip surface was regenerated with 50 mM NaOH and reloaded with Sus11-IgE.

To determine binding kinetics of ligelizumab and omalizumab IgG for wild-type C328 and mutated C335 IgE-Fc$_{3-4}$ variants, 100 RU of ligelizumab or omalizumab IgG were immobilized on individual chips (ligelizumab: pH 5.0, omalizumab: pH 4.5). A blank immobilization was performed on flow cell 1. Different concentrations (0.3–5 nM) of C328 and C335 IgE-Fc$_{3-4}$ were injected for 120 s and the dissociation was measured for 180 s under constant buffer flow. After each run, the chip surface was regenerated with 10 mM glycine-HCl pH 2.0.

To assess whether ligelizumab or omalizumab IgG recognize FcεRIα:IgE complexes, 1000 RU of recombinant human FcεRIα was immobilized on flow cell 2 at pH 4.0. A blank immobilization was performed on flow cell 1. A concentration of 20 nM Sus11-IgE was subsequently captured for 120 s to reach a response of >100 RU. Various concentrations (3.13–100 nM) of ligelizumab, omalizumab antibodies/fragments and Le27 were injected for 120 s and the dissociation was measured for 180 s under constant buffer flow. After each run, the chip surface was regenerated with 50 mM NaOH and reloaded with Sus11-IgE.

To assess whether ligelizumab or omalizumab IgG may disrupt FcεRIα:IgE complexes, 1000 RU of recombinant human FcεRIα was immobilized on flow cell 2 at pH 4.0. A blank immobilization was performed on flow cell 1. A concentration of 20 nM Sus11-IgE was subsequently captured for 120 s to reach a response > 100 RU. Three concentrations (0.25, 0.5 and 1 μM) of ligelizumab or omalizumab antibodies/fragments were injected for 42 times 540 s with a dissociation time of 180 s between each injection under constant buffer flow. At the end of each run the chip surface was regenerated with 50 mM NaOH and reloaded with Sus11-IgE.

To assess whether ligelizumab or omalizumab IgG may disrupt FcεRIα:IgE-Fc$_{3-4}$ complexes, 1000 RU of recombinant human FcεRIα was immobilized on flow cell 2 at pH 4.0. A blank immobilization was performed on flow cell 1. A concentration of 20 nM IgE-Fc$_{3-4}$ was subsequently captured for 120 s to reach a response > 100 RU. Various concentrations (3.13–100 nM) of ligelizumab, omalizumab antibodies/fragments and Le27 were injected for 120 s and the dissociation was measured for 180 s under constant buffer flow. After each run, the chip surface was regenerated with 50 mM NaOH and reloaded with IgE-Fc$_{3-4}$.

To assess whether ligelizumab or omalizumab IgG interact with CD23:IgE complexes, 3000 RU of recombinant human CD23 was immobilized on flow cell 2 at pH 4.0. A blank immobilization was performed on flow cell 1. A concentration of 125 nM JW8-IgE was captured for 120 s to reach a response > 100 RU. Various concentrations (31.25–500 nM) of ligelizumab, omalizumab antibodies were injected for 120 s and the dissociation was measured for 180 s under constant buffer flow. After each run, the chip surface was regenerated with 50 mM NaOH and reloaded with JW8-IgE. Ligelizumab and omalizumab IgG binding to preformed CD23:IgE complexes was assessed using a polyclonal sheep anti-human IgG antibody at 125 nM concentration (The Binding Site, Birmingham, UK).

**Protein interaction measurements with ELISA.** To measure the binding interaction of ligelizumab or omalizumab with human IgE, the anti-IgE antibodies were immobilized on plastic surface of a 96-half-well plate (Corning, NY, USA) at a concentration of 30 nM by overnight incubation in PBS at 4 °C. The next day, the plate was blocked with PBS/0.15% casein for 2 h at room temperature (RT) and washed with PBS/0.05% Tween. Human Sus11-IgE was incubated at a serial dilution (0.03–7.5 nM) for 1 h at RT. The plate was washed two times with PBS/0.05% Tween and incubated for 1 h with the same wash buffer followed by three times washing with PBS only. IgE was detected with monoclonal non-competitive

anti-IgE Le27 coupled to horseradish-peroxidase (HRP). TMB (3,3′,5,5′-tetra-methylbenzidine, Merck, Darmstadt, Germany) was used as a substrate for HRP and the reaction was stopped with 1 M sulfuric acid. Absorbance was measured at 450 nm wavelength using the standard ELISA reader SpectraMax M5 from Molecular Device LLC (San Jose, CA, USA).

To measure the binding interaction of ligelizumab or omalizumab with wild-type C328 and mutated C335 IgE-Fc$_{3-4}$ variants, the anti-IgE antibodies were immobilized on plastic surface of a 96-half-well plate (Corning, NY, USA) at a concentration of 10 nM by overnight incubation in PBS at 4 °C. The next day, the plate was blocked with PBS/0.15% casein for 2 h at RT and washed with PBS/0.05% Tween. Wild-type C328 and mutated C335 IgE-Fc$_{3-4}$ variants were incubated at a serial dilution (0.0001–10 nM) for 1 h at RT and subsequently washed with PBS/0.05% Tween and PBS. Development of the assay was performed as mentioned above.

To measure IgE-binding to FcεRI, we immobilized recombinant human FcεRIα on a plastic surface of a 96-half-well plate (Corning, NY, USA) at 30 nM by overnight incubation in PBS at 4 °C. The next day, the plate was blocked with PBS/0.15% casein for 2 h at RT. Biotinylated JW8-IgE was incubatd at a 1:3 serial dilution (60–0.001 nM) for 1 h at RT. IgE-binding was detected with poly-HRP-conjugated streptavidin (Thermo Scientific, Waltham, MA, USA) and was followed by development with TMB.

To assess the inhibition of IgE-binding to FcεRIα by ligelizumab and omalizumab, we immobilized recombinant human FcεRIα on plastic surface of a 96-half-well plate (Corning, NY, USA) at a concentration of 30 nM by overnight incubation in PBS at 4 °C. The next day, the plate was blocked with PBS/0.15% casein. Anti-IgE antibodies (0.0078–600 nM) were pre-incubated with biotinylated JW8-IgE (0.78 nM) for 30 min at RT and were added for 1 h to the plate. The plate was washed with PBS/0.05% Tween and PBS. IgE binding was detected with poly-HRP-conjugated streptavidin and was followed by development with TMB.

For the CD23-binding ELISA, we immobilized 60 nM recombinant human CD23 (R&D Systems, Minneapolis, MA, USA) on a 96-half-well plate (Corning, NY, USA) by overnight incubation at 4 °C. The following day, the plate was blocked with PBS/0.15% casein. The JW8-IgE was mixed 1:1 with NIP$_7$-BSA (Biosearch Technologies, Petaluma, CA, USA) in PBS/0.15% casein at a 1:2 serial dilution (280–2.1875 nM) and incubated for 30 min at RT. The IgE-antigen complexes were then incubated on the plate for 1 h at RT. IgE binding was detected with biotinylated anti-IgE antibody Le27 (2.66 µg/ml) and poly-HRP-conjugated streptavidin followed by development with TMB. In the inhibition ELISA for CD23-binding IgE-antigen complexes (70 nM:70 nM) were incubated with a 1:2 serial dilution of the anti-IgE antibodies (280–0.5469 nM) for 30 min at RT before incubation on the ELISA plate.

**Protein interaction measurements on cells.** Primary human basophils and BDCA1$^+$ DCs were isolated from whole-blood donations. Human peripheral whole-blood was obtained from volunteering donors, who provided informed consent in accordance with the Helsinki Declaration. The study was approved by the local ethics committee (KEK 2018-00204). Basophils and BDCA1$^+$ DCs were enriched by Percoll density centrifugation of dextran-sedimented supernatants. Furthermore, basophils were purified with negative selection using the Milteny basophil isolation kit II (Miltenyi Biotec, Bergisch Gladbach, Germany). BDCA1$^+$ DCs were isolated with positive selection using the Milteny human CD1s (BDCA-1$^+$ dendritic cell isolation kit (Miltenyi Biotec, Bergisch)). Cells were analyzed for purity via flow cytometry.

To compare the efficacy of ligelizumab and omalizumab IgG to inhibit IgE binding to FcεRIα expressing cells, we cultured isolated primary human basophils at a density of $1 \times 10^6$ cells/ml per well in a 96-well round-bottom plate (Falcon, Tewksbury, MA, USA). First, we removed endogenous IgE from the cell surface by addition of 5 µM of the disruptive anti-IgE DARPin® protein bi53_79 in RPMI$^{+/+}$ supplemented with 10 ng/ml recombinant human IL-3 overnight for basophils. The next day, the cells were washed and incubated with different concentrations of JW8-IgE (0.006–100 nM) to assess dose-dependent binding. The concentration of 2 nM was further used for inhibition experiments with anti-IgE antibodies. JW8-IgE was pre-incubated with increasing concentrations (0.032–100 nM) of ligelizumab or omalizumab for 30 min at RT. Surface IgE was stained and quantified by flow cytometry 1 h after adding this mix to the cells. To assess the inhibition profile of ligelizumab and omalizumab IgG to inhibit IgE binding to FcεRIα BDCA-1$^+$ DCs without the presence of any additionally cytokines or growth factors, we modified the protocol above and removed the endogenous IgE from the cells by the disruptive anti-IgE DARPin® protein bi53_79 only in RPMI$^{+/+}$ medium during an incubation of 4 h. Subsequently, the cells were washed and incubated with different concentrations of JW8-IgE (100–0.005 nM) to assess dose-dependent binding. The concentration of 25 nM JW8–IgE was further used for inhibition experiments with anti-IgE antibodies. JW8-IgE was pre-incubated with increasing concentrations (0.0012–312.5 nM) of ligelizumab or omalizumab for 30 min at RT. Surface IgE was stained and quantified by flow cytometry 1 h after adding this mix to the cells. Surface IgE was stained and quantified by flow cytometry 1 h after adding this mix to the cells.

To assess inhibition of IgE-dependent basophil activation, we first determined the concentration of JW8-IgE to be used to reach half-maximal activation. For this we titrated JW8-IgE (0.0064–20 nM) on 25,000 anti-IgE DARPin® protein-treated

human primary basophils and subsequently stimulated these cells with 100 ng/mL NIP$_7$-BSA (Biosearch Technologies, Petaluma, CA, USA) in RPMI$^{+/+}$-containing human 10 ng/ml recombinant human IL-3 and Flow CAST® kit antibody staining mix (Bühlmann Laboratories AG, Schönenbuch, CH). The concentration of 0.68 nM JW8-IgE was subsequently used for pre-incubation with increasing concentrations (0.0064–20 nM) of ligelizumab and omalizumab in RPMI$^{+/+}$ medium for 30 min at RT. Anti-IgE DARPin® protein-treated primary human basophils were then sensitized for 2 h at 37 °C with this mixture and stimulated with 100 ng/mL NIP$_7$-BSA. Activation was determined by measuring CD63$^+$ basophils using flow cytometry.

To investigate whether anti-IgE antibodies might actively remove FcεRIα-bound IgE from the cell surface, we cultured 50,000 isolated primary human basophils ($1 \times 10^6$ cells/ml) with either 0, 0.25, 0.5 or 1 µM of ligelizumab or omalizumab antibodies/fragments in RPMI$^{+/+}$ supplemented with 10 ng/ml recombinant human IL-3. After 3 and 6 days of culture, the cells were stained with anti-IgE antibody for 15 min at RT and measured by flow cytometry.

The human RPMI8866 leukemia B-cell line was kindly provided by Dr. Monique Vogel. The cells were cultured in RPMI$^{+/+}$ medium at a density of $2.5 \times 10^5$ cells/ml in a 250-ml cell culture flask (Greiner Bio One, Kremsmünster, AUT). One day before the experiment, the cells were split 1:2 in RPMI$^{+/+}$ medium. Flow cytometry was performed using a BD FACSCanto device (BD Bioscience, Franklin Lakes, NJ, USA) and results were evaluated with FlowJo Version 10.1 (Ashland, OR, USA).

To assess whether ligelizumab accelerates dissociation of FcεRI-bound IgE-Fc$_{3-4}$ on purified primary human basophils (purity > 90%), endogenous IgE was removed from 50,000 basophils per well by incubation with 5 µM disruptive anti-IgE DARPin® protein bi53_79 in RPMI$^{+/+}$. After washing with PBS pH 7.4, the cells were reloaded with 100 nM JW8-IgE or C328 IgE-Fc$_{3-4}$. Again the cells were washed with PBS pH 7.4. Then the cells were treated with 0.6–66.6 nM ligelizumab or omalizumab IgG in RPMI$^{+/+}$ medium containing human 10 ng/ml recombinant human IL-3 for 30 min at 37 °C and 5 % CO$_2$. Surface expression of human IgE was determined with an anti-human IgE FITC antibody and basophil activation was assessed using the Flow CAST® kit (Bühlmann Laboratories AG) by flow cytometry.

For IgE titration on CD23 expressing cells, $5 \times 10^4$ RPMI8866 cells per well were seeded in a 96-well plate (Corning, NY, USA) and incubated with biotinylated JW8-IgE for 1 h at 37 °C, 5% CO$_2$ in RPMI$^{+/+}$ medium. Subsequently, the cells were stained with streptavidin FITC (Invitrogen, Carlsbad, CA, USA), anti-CD23 and anti-CD19 antibodies for 20 min at 4 °C. For the IgE inhibition on CD23 expressing cells, 12.5 nM biotinylated JW8-IgE was pre-complexed with the anti-IgE antibodies (0.01–25 nM, 1:2 serial dilution) in RPMI$^{+/+}$ medium for 30 min at RT. Subsequently, this mix was added to the RPMI8866 cells for 1 h at 37 °C, 5% CO$_2$ in RPMI$^{+/+}$ medium. Staining was performed as mentioned above.

To check for binding of anti-IgE antibodies to CD23-bound IgE RPMI8866 cells were incubated with 12.5 nM biotinylated JW8-IgE for 1 h at 37 °C, 5% CO$_2$ in RPMI$^{+/+}$ medium. The cells were then washed three times with 150 µl of PBS pH 7.4 and centrifuged at $600 \times g$ for 5 min at 4 °C, resuspended in RPMI$^{+/+}$ medium containing 12.5 nM ligelizumab, omalizumab or control IgG and incubated for 30 min at 37 °C. Subsequently, the cells were washed two times with 200 µl of PBS pH 7.4 and stained with streptavidin, anti-CD19 and monoclonal mouse anti-human Ig κ light chain (clone TB28-2, Thermo Fisher Scientific, MA, USA) at 4 °C. Blocking of CD23 receptor was performed by incubation of the RPMI8866 cells with 5 nM anti-CD23 DARPin® protein D89_86 (ref. [44]) for 15 min at 37 °C, 5% CO$_2$. Cells were then washed once with PBS before incubation with JW8-IgE and the anti-IgE antibodies as described above.

To assess the effect of ligelizumab and omalizumab on IgE:antigen complex internalization with BDCA1$^+$ DCs, we conjugated pH-sensitive rodamine (pH-Rodo) to NIP-specific JW8 IgE by the pHrodo Red Microscale Labeling Kit (Life Technologies). The fluorophore pH-Rodo increases emission at more acidic pH, which happens along the pathway of endocytosis/phagocytosis (i.e. endosomal/lysosomal pH range from pH 6 to 4.5). First pH-Rodo IgE:antigen complexes were formulated with 25 nM pH-rodo IgE and 2.5 nM NIP(15)BSA (Biosearch Technologies, Petaluma, California, USA) for 0.5 h at RT and were then supplemented with an 5 nM ligelizumab or omalizumab for additional 0.5 h at RT. Upon removing the endogenous IgE from basophils and BDCA1$^+$ DCs by the disruptive anti-IgE DARPin® protein bi53_79 for 4 h at 37 °C, the cells were washed and pulsed with formulated IgE:antigen:anti-IgE antigen complexes for 2 h at 37 °C. After pulsing the cells, the cells were washed intensively and further incubated time-dependently (i.e. 4 h, 8 h, 12 h, 16 h) at 37 °C. The pH-Rodo JW8-IgE emissions were finally measured in the PE-channel by flow-cytometry.

To assess the binding of anti-IgE antibodies to CD23-bound IgE by multispectral imaging flow cytometry, RPMI8866 cells were diluted to $1 \times 10^6$ cells/ml, and $1 \times 10^5$ cells per well were seeded in a 96-well round-bottom plate. Washing was performed once with 200 µl of PBS pH 7.4 at $500 \times g$ for 5 min at 4 °C. The cells were then resuspended in 100 µl of RPMI$^{+/+}$ medium containing 12.5 nM biotinylated JW8-IgE and incubated for 1 h at 37 °C, 5% CO$_2$. Afterward, the cells were washed three times with 200 µl of PBS pH 7.4 at $500 \times g$ for 5 min at 4 °C, followed by 30 min incubation at 37 °C, 5% CO$_2$ in 100 µl of RPMI$^{+/+}$ medium containing 12.5 nM ligelizumab, omalizumab or isotype IgG. Subsequently, the cells were stained with anti-CD23, anti-Ig κ light chain and streptavidin antibodies for 20 min at 4 °C. Binding of IgE and the anti-IgE

antibodies was assessed using an Amnis® ImageStream®X MKII and the corresponding IDEAS® software (Luminex corporation, Austin, TX, USA).

**IgE ELISpot and culture supernatants**. Peripheral blood mononuclear cells were isolated from fresh whole blood supplemented with 100 mM EDTA using Ficoll Paque PLUS (GE Healthcare, Chicago, IL, USA) density gradient centrifugation. The blood was diluted 1:2 in PBMC wash buffer (PBS pH 7.4 complemented with 2% FCS, 2 mM EDTA), overlaid onto Ficoll layer and centrifuged at 1800 rpm for 35 min at RT with brakes off. The peripheral blood leukocyte (PBL) layer was extracted and washed with PBMC wash buffer at 1200 rpm for 5 min, followed by two times washing with PBS pH 7.4 at 1000 rpm for 10 min.

A total of $2.5 \times 10^5$ PBMCs were cultured in 96-well round-bottom plates at a density of $1 \times 10^6$ cells/ml in RPMI$^{+/+}$ medium. The cells were stimulated with 30 ng/ml recombinant human IL 4 and 1 μg/ml anti-human CD40 antibody for 6 days at 37 °C, 5% $CO_2$ in the presence of 0.5 μM anti-IgE antibodies. Unstimulated, untreated and monoclonal anti-human IgG1 antibody-treated cells served as controls. For the detection of IgE-producing B-cells, we used 96-well MultiScreen filter plates (Merck Milipore, Burlington, MA, USA) and the human IgE ELISpotBASIC kit (Mabtech, Nacka Strand, Sweden). Overnight incubation of the coating antibody (15 μg/ml) at 4 °C was followed by blocking for 2 h with RPMI$^{+/+}$ medium the next day. The PBMCs were washed twice with PBS pH 7.4 before transfer to a MultiScreen plate and incubation in RPMI$^{+/+}$ medium complemented with recombinant human IL-4 (30 ng/ml) and anti-CD40 (1 μg/ml) for 24 hours at 37 °C, 5% $CO_2$. For IgE detection, Mabtech detection antibody mixture (1 μg/ml, 5 h) and poly-HRP-conjugated streptavidin (0.17 μg/ml, 30 min) were diluted in RPMI$^{+/+}$ medium and incubated on ELISA plate shaker at RT. For HRP substrate, we used the AEC staining kit (Sigma, Kawasaki, Kanagawa, Japan) and incubated the plate with 50 μl per well for 5 min. The colorimetric reaction was stopped by washing the plate with ddH2O. Washing steps were performed with PBS pH 7.4 or PBS pH 7.4/0.05% Tween. Spots were quantified using an ELISpot reader and the corresponding iSpot software v7.0 (AID GmbH, Strassberg, Germany).

To assess the effect of anti-IgE antibodies on the production of soluble IgE, we cultured PBMCs at $1 \times 10^6$ cells/ml in RPMI$^{+/+}$ medium at 37 °C, 5% $CO_2$ in the presence of 0.5 μM anti-IgE antibodies and 0.5 μM of the anti-FcγRIIB DARPin® protein D11 (ref. [56]), respectively. IgE production was induced by stimulation with 30 ng/ml recombinant human IL-4 and 1 μg/ml anti-human CD40 antibody for 12 days. On day 12, the supernatants were collected and soluble IgE was measured by ELISA. For the ELISA, anti-IgE antibody Le27 was immobilized on plastic surface of a 96-half-well plate (Corning, NY, USA) at a concentration of 30 nM by overnight incubation in PBS at 4 °C. The next day, the plate was blocked with PBS/0.15% casein. The culture supernatants were incubated for 1 h on the plate followed by washing twice with PBS pH 7.4 or PBS pH 7.4/0.05% Tween. IgE was detected using Mabtech detection antibody mixture (1 μg/ml, 1 hour) and poly-HRP-conjugated streptavidin (0.17 μg/ml, 30 min) diluted in PBS/0.15% casein and incubated on ELISA plate shaker at RT. TMB (Merck, Darmstadt, Germany) was used as a substrate for HRP and the reaction was stopped with 1 M sulfuric acid. Absorbance was measured at 450 nm wavelength using the standard ELISA reader SpectraMax M5 from Molecular Device LLC (San Jose, CA, USA).

**In vivo passive systemic anaphylaxis**. Mice transgenic for human FcεRIα (huFcεRIα tg) on a mixed C57BL/6J–C57BL/6N background were obtained from Prof. J.-P. Kinet. All animal experimentation was approved by the local ethics committee (authorization BE66/18). Seven days before the experiment, huFcεRIα tg mice were subcutaneously implanted with an electronic temperature transponder (IPTT-300) from BMDS (Delaware, USA) to measure body-core temperature as instructed by the manufacturer. On day 0, mice received an intraperitoneal injection of PBS or 20 μg of anti-IgE antibody (ligelizumab or omalizumab in 200 μl). Half an hour later, mice were passively sensitized with 20 μg of NIP-specific human JW8-IgE (in 200 μl) by intraperitoneal injection. On day 1, mice were challenged by intraperitoneal injection of 200 μg of NIP$_{20}$-BSA (Biosearch Technologies, Petaluma, CA, USA). Body-core temperature was measured before challenge (baseline) and every 10 min after antigen-challenge for 60 min and every 30 min until 120 min. Data are represented as measured temperature after challenge minus baseline temperature (Δ core body temperature) for each time point.

**Reporting summary**. Further information on research design is available in the Nature Research Reporting Summary linked to this article.

## Data availability
Coordinates of the IgE-Fc:ligelizumab-scFv complex structure have been deposited to the Protein Data Bank (PDB) under accession number 6UQR (https://doi.org/10.2210/pdb6uqr/pdb). Structural information about FcεRIα:IgE-Fc$_{2-4}$ (PDB ID: 2Y7Q (https://doi.org/10.2210/pdb2Y7Q/pdb)[53], CD23:IgE-Fc$_{2-4}$ (PDB ID: 4EZM (https://doi.org/10.2210/pdb4EZM/pdb)[57] and Omalizumab-Fab:IgE-Fc$_{3-4}$ (PDB ID: 5HYS (https://doi.org/10.2210/pdb5HYS/pdb)[41]) complexes are accessible in the Protein Data Bank (PDB). Raw data for Figs. 1, 2, 3, 4, 5, and Supplementary Figs. 1, 2, 3, 5 are provided in the Source Data file. All other data are available from the corresponding authors upon request.

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

## Acknowledgements

We thank all members of the Eggel and Jardetzky labs involved in this study. We further acknowledge Dr. Mauro Zurini for the preparation and analysis of the anti-IgE antibody fragments, Prof. Jean-Pierre Kinet for providing the transgenic mice expressing the human FcεRIα, Bühlmann Laboratories AG for scientific support, Dr. Monique Vogel and Prof. Martin Bachmann for providing cell lines and granting access to equipment, Dr. Paul Engeroff for technical and experimental support and Prof. Peter M. Villiger, and Dr. Reinhold Janocha and Dr. Maximilian Woisetschlaeger for discussions and research support. This research was funded by a grant from the Fondation Acteria (to A.E.), a Swiss National Science Foundation Ambizione grant PZ00P3_148185 (to A.E.), the Research Fund of the Swiss Lung Association, Bern and the Uniscientia foundation (to A.E.), consumable contributions by Novartis AG (to A.E.), and NIH grants AI115469 (to T.S.J.) and HL141493 (to T.S.J. and A.E.).

## Author contributions

C.H., T.S.J. and A.E. conceptualized the study and wrote the manuscript, P.Ga., S.S.T., P. Gu., D.B., R.R., N.Z., S.K. and A.E. performed experiments. All authors contributed to data interpretation and discussion.

## Conflict of interest

C.H. and A.E. are consulting for Novartis Pharma AG. All other authors declare no competing interests.
