## [Peer Review File · Nature Communications]

Reviewers' comments:

Reviewer #1 (IgE response, germinal center reaction)(Remarks to the Author):

Manuscript Number: 19-15207

Title: The next generation anti-IgE antibody ligelizumab efficiently suppresses FcεRI-dependent allergic responses through neutralization of free IgE and inhibition of IgE production

Authors: Pascal Gasser^{1,2,3}, Svetlana S. Tarchevskaya⁴, Pascal Guntern^{1,2}, Daniel Brigger^{1,2}, Noemi Zbären^{1,2}, Silke Kleinboelting⁴, Christoph Heusser⁵, Theodore S. Jardetzky⁴, and Alexander Eggel^{1,2}

The authors investigated the anti-IgE antibody, ligelizumab, a novel high-affinity monoclonal anti-IgE antibody that has been developed for patients with allergic disorders. They determined the molecular binding profile and show that ligelizumab inhibits effectively IgE binding to FcεRI and less to CD23. The crystal structure of ligelizumab bound to IgE revealed differences in its binding epitope compared to omalizumab.

The manuscript is well-written and follows from their previous publication [Pennington Nat Commun 2016] in which the molecular characterization and therapeutic efficacy of the other clinical anti-IgE antibody, Omalizumab, was reported. In the current paper, they show that Ligelizumab is superior in inhibition of IgE binding to FcεRI, basophil activation, IgE production by B cells and passive systemic anaphylaxis in an *in vivo* mouse model. What is lacking in the paper is the characterization of the mechanism underlying ligelizumab-induced suppression of IgE synthesis.

The following questions should be addressed:

- Their findings on ligelizumab-induced IgE binding to FcεRI and CD23 (Fig. 2) should be related to physiological concentrations of IgE found in allergic patients. In other words, are the IgE concentrations used in their experiments in line with IgE levels found in patients? Along the same line, are the concentrations of ligelizumab used in their *in vitro* experiments (Fig. 2,3,4) in line with concentrations observed in allergic patients treated with anti-IgE antibodies?
- Basophils are not the only FcεRI-expressing cells in human. Monocytes and BDCA3⁺ dendritic cells also express FcεRI at the cell surface [A. Greer JCI 2014]. The effect of ligelizumab on IgE binding to FcεRI should also be investigated on these immune cell types (which can be easily purified from human peripheral blood) and the functional effects of ligelizumab on cell activation should be studied.
- FcεRI internalization essays should be performed to analyze what the effect is of ligelizumab on FcεRI-IgE recycling in live cells (basophils/monocytes) cultured in time.
- What is the underlying mechanism of ligelizumab-induced suppression of IgE synthesis (Figure 4)? It is speculated in the discussion that antibody-dependent cellular cytotoxicity (ADCC) of B-cells may be involved, but also a direct effect via CD23 stabilization or cross-linking is possible. They should perform experiments to distinguish between these different possibilities.
- In the mouse model, there is no human CD23 present. If this is the reason why Ligelizumab is more effective than Omalizumab *in vivo*, then the question is how relevant this finding is. Have the authors performed other *in vivo* studies (other allergic (humanized) mouse models) to exclude this?

Other points:

- The reported affinity (Kd) of omalizumab in Table 1 is 10x times higher than previously reported work [E. Cohen mAbs 2014]. Can the authors comment on this?
- For most experiments, it is not clearly stated how many independent experiments were performed. Please include this information in the legend, including supplementary legends.
- It seems that the in vivo experiment (Fig. 4) has been performed once with only 3 mice/group??

Reviewer #2 (Immune structure, FcR)(Remarks to the Author):

IgE antibodies and its high affinity receptor Fc ϵ RI are the fundamental components of allergic response, and their interaction leads to immediate hypersensitivity reactions against various allergens. In addition to high affinity receptor Fc ϵ RI, IgE can also bind to another receptor on mast cells, namely CD23. Unlike the high affinity receptor Fc ϵ RI that is mainly responsible for IgE-mediated mediator release, CD23 might have both positive and negative regulation effects on IgE production. The investigation of the interaction between IgE and its receptors has been a fundamental question in structural immunology. In the meanwhile, the advance in the understanding of these structural mechanisms also supports the therapeutic strategy to block the IgE-IgE receptor interaction to alleviate allergic response.

For example, therapeutic antibody, Omalizumab, significantly decreases serum IgE in patients and has been approved for clinical treatments for allergic asthma. However, this type of blocking antibodies prevents IgE from its binding to both Fc ϵ RI and CD23 and thus undermines the potential beneficial effect of CD23. Therefore, there is a unique request to raise antibodies to differentially block IgE binding to its high affinity and low affinity receptors. Toward this goal, a new high affinity anti-IgE monoclonal antibody (ligelizumab; QGE031) showed greater efficacy than Omalizumab on inhaled and skin allergen responses in patients with mild allergic asthma, representing a promising next generation therapeutic antibody to suppress Fc ϵ RI-mediated allergic response.

In the current manuscript, the authors characterized the binding of ligelizumab to IgE and further determined its binding epitope on IgE using X-ray crystallography. Despite the intermediate resolution, their results revealed that ligelizumab binds to free IgE at sites with a major overlapping with Fc ϵ RI and a minor overlapping with CD23. However, ligelizumab was not able to displace cell-surface bound IgE like Omalizumab. Instead, it could bind to IgE:CD23 on the cell surface, suggesting a unique process to mediate its therapeutic effects. Although the characterization of the binding of ligelizumab to IgE in the presence or absence of Fc ϵ RI was well carried out, the binding of ligelizumab to IgE:CD23 complexes in solution or on the cell surface was not complete and the physiological effect of such binding needs to be further investigated to establish a solid foundation for understanding the highly efficient suppression of Fc ϵ RI-mediated allergies. Therefore, a revision is necessary before the current manuscript could be published in the journal of Nature Communication to shed new lights in the field to further develop this class of antibodies toward better therapeutic effects.

1) In Fig2, the authors showed that both Ligelizumab can block the binding of IgE to both Fc ϵ RI and CD23, which agrees with their structure observation that Ligelizumab's binding epitope on IgE is overlapping with those of Fc ϵ RI and CD23, suggesting mutually exclusive bindings. Furthermore, the BIAcore studies in Fig 3 confirmed that Ligelizumab can't bind to and dissociate IgE bound onto Fc ϵ RI. A similar BIAcore study should be carried out to demonstrate the binding among IgE, CD23, and Ligelizumab to support the idea that Ligelizumab could bind to IgE:CD23 complexes.

2) In Fig4e and Fig4f, the MFI of cell surface IgE in the presence of Ligelizumab was ~5000 and 750, respectively. If the data were obtained in the same experiment, it suggests that a very small portion of cells could be stained with streptavidin(for IgE) and anti-human Ig light chain (for Ligelizumab).

Therefore, the authors' conclusion that Ligelizumab was able to bind to IgE:CD23 complex is shaky. The detailed FACS staining dot profile should be provided. In addition, a microscopic investigation should be carried out to provide further information to support authors' proposal.

Reviewer #3 (Antibody response, immune structure)(Remarks to the Author):

The authors have used crystallography, SPR, ELISA, various cell-based in vitro assays, and a mouse model to compare the activities and modes of action of two anti-IgE antibodies of interest for suppression of allergic responses in humans, omalizumab and ligelizumab. The authors distinguish allergy signalling through two cell surface receptors, Fc_epsilon_RI and CD23. Their data demonstrate that ligelizumab has higher affinity and avidity for IgE than omalizumab and prefers the open conformation of IgE C-epsilon3, whereas omalizumab binds similarly to both open and closed conformations. Their crystal structure of ligelizumab ScFv bound to an IgE Fc3-4 fragment reveals the binding mode, shows how it differs from that of omalizumab, and is consistent with ligelizumab favoring a more open conformation. ELISA and flow cytometry showed that ligelizumab blocked IgE binding to Fc-epsilon-RI more efficiently than omalizumab, but in contrast, ligelizumab was inferior to omalizumab for blocking IgE binding to CD23. The authors postulated structural mechanisms for this difference in specificity, based on analysis of their crystal structure and others in the literature. The authors showed that the two antibodies act differently in other potentially important ways too: (i) omalizumab but not ligelizumab can accelerate dissociation of IgE from the surface of FC-epsilon-RI-expressing cells; (ii) while omalizumab binds to preformed complexes of FC-epsilon-RI with IgE Fc3-4, ligelizumab disrupts such complexes; (iii) ligelizumab but not omalizumab removes cell surface IgE-Fc3-4 in a dose-dependent manner; (iv) omalizumab but not ligelizumab activates basophils re-sensitized with IgE-Fc3-4 in a dose-dependent manner. Finally, in terms of functional efficacy, the authors showed that ligelizumab has higher potency in inhibiting basophil activation; provided B cell ELISpot evidence that ligelizumab is superior to omalizumab for suppression of IgE production by B cells; and showed that ligelizumab is superior for protection against anaphylaxis in a mouse model.

The work addresses questions of significant relevance to treatment of human allergic diseases. The manuscript is well-written, the conclusions are carefully drawn and seem well supported by the data. The work provides a solid pre-clinical foundation on which to assess clinical studies of ligelizumab, and potentially opens the door to future therapies that further optimize the parameters explored here. Areas of further exploration would certainly include the distinctions noted in interfering with Fc_epsilon_RI vs CD23 signalling and the potential that omalizumab cross-linking of complexes of Fc-epsilon-RI and IgE-Fc3-4 may be related to the observed low level of anaphylaxis of omalizumab-treated patients, whereas ligelizumab may avoid that complication by not binding to such complexes if they do in fact exist in vivo.

Minor

page 3. binding of IgE to FCepsilonRI is stated to have high affinity of KD 0.1 nM. Is this a monovalent KD? Please clarify if bivalent avidity is in play here.

page 3. "...cross-linking of FCeRI-bound IgE..." should that not be "...cross-linking of IgE-bound FCeRI..."?

page 5. A KD of 18 pM is given for the interaction of ligelizumab with IgE. The format of the SPR measurement suggests that bivalent avidity is in play here. Please clarify.

page 5. The affinity of the ligelizumab Fab fragment for IgE is given as 35 nM, which suggests to this reviewer that the 18 pM KD given for the full ligelizumab IgG indeed includes the effect of bivalent avidity. Yet, the authors say "The affinity of the ligelizumab Fab fragment was 35 nM, indicating that intact antibody binding was not avidity driven." This last statement seems incorrect. Please clarify.

page 9. when comparing the binding modes of ligelizumab/IgE-Fc and CD23/IgE-Fc, the authors distinguish between "steric overlap" and "competition for IgE surface subsites". But these are not mutually exclusive concepts -- in order for CD23 and ligelizumab to bind to the same subsites on IgE, some steric overlap would obviously occur.

First, we would like to thank the editor and the three referees for dedicating their time to review our submitted manuscript so carefully. The comments were well received and appreciated. We value that the editors and referees overall share our judgement that this study and its results will be of relevance for the readers of Nature Communications. In the meantime, we have carefully considered all remarks made by the referees and have conducted additional experiments to specifically address the various points. We are convinced that the provided expert opinion has further improved the quality of this study. Please find below a point-by-point reply to the referee's comments (in blue).

Reviewer #1 (IgE response, germinal center reaction)(Remarks to the Author):

The authors investigated the anti-IgE antibody, ligelizumab, a novel high-affinity monoclonal anti-IgE antibody that has been developed for patients with allergic disorders. They determined the molecular binding profile and show that ligelizumab inhibits effectively IgE binding to FcεRI and less to CD23. The crystal structure of ligelizumab bound to IgE revealed differences in its binding epitope compared to omalizumab.

The manuscript is well-written and follows from their previous publication [Pennington Nat Commun 2016] in which the molecular characterization and therapeutic efficacy of the other clinical anti-IgE antibody, Omalizumab, was reported. In the current paper, they show that Ligelizumab is superior in inhibition of IgE binding to FcεRI, basophil activation, IgE production by B cells and passive systemic anaphylaxis in an in vivo mouse model. What is lacking in the paper is the characterization of the mechanism underlying ligelizumab-induced suppression of IgE synthesis.

The following questions should be addressed:

R1-Comment-1: Their findings on ligelizumab-induced IgE binding to FcεRI and CD23 (Fig. 2) should be related to physiological concentrations of IgE found in allergic patients. In other words, are the IgE concentrations used in their experiments in line with IgE levels found in patients? Along the same line, are the concentrations of ligelizumab used in their in vitro experiments (Fig. 2,3,4) in line with concentrations observed in allergic patients treated with anti-IgE antibodies?

We apologize for not being clear on this but what is shown in the original Fig. 2a-d is ligelizumab-mediated inhibition of and not ligelizumab-induced IgE-binding to FcεRIα and CD23. We would like to point out that the physiological concentrations of total IgE in allergic patients vary considerably (range: 0.2-2500 IU/ml)¹ - thus by a factor of up to 10⁴ - and it is widely accepted in the field that serum IgE levels do not correlate with or predict disease severity². Moreover, there is no specific threshold value that separates patients with allergic disease from those without. For the in vitro inhibition experiments shown in Fig. 2 we performed IgE titrations (original Supplementary Fig. 2) to find the optimal dose to be used in each of these assays as it is standard laboratory practice for such biophysical and enzyme immunoassays. The IgE concentration to be used in inhibition assays is greatly dependent on the amount of receptor that is immobilized on the solid phase. For most of the assays we used IgE concentrations in the low nanomolar range, which would correspond to IgE serum level around 100 IU/ml in a patient. Thus, the concentrations used in the assays are indeed physiologically relevant.

The amount of ligelizumab used ranges from low picomolar to the low micromolar concentrations. Wherever possible, we included titration curves covering a broad range of concentrations. Pharmacokinetic studies with ligelizumab in human patients have revealed serum concentrations of ~1-100 $\mu\text{g}/\text{ml}$ corresponding to 6-600 nM^3 . Therefore, the anti-IgE concentrations used in the assays also fall within the physiologically relevant range corresponding to exposure concentrations in patients during treatment. We have revised the text accordingly (pg.7, ln.138; pg.7, ln.144; pg.8, ln.152; pg.8, ln.158).

R1-Comment-2: Basophils are not the only Fc ϵ RI-expressing cells in human. Monocytes and BDCA3+ dendritic cells also express Fc ϵ RI at the cell surface [A. Greer JCI 2014]. The effect of ligelizumab on IgE binding to Fc ϵ RI should also be investigated on these immune cell types (which can be easily purified from human peripheral blood) and the functional effects of ligelizumab on cell activation should be studied.

*It is our understanding that BDCA3+ dendritic cells do not express Fc ϵ RI α ⁴, but we assume that the reviewer is referring to BDCA1+ dendritic cells (DCs) that were shown to display Fc ϵ RI α on their surface⁵. To address the reviewer's comment, we isolated basophils, BDCA1+ dendritic cells and monocytes from human whole blood donations (**Response Fig. 1a**) and first assessed Fc ϵ RI α , CD23 as well as the different IgG-receptor expression by flow cytometry (**Response Fig. 1b**). Basophils clearly display the highest levels of Fc ϵ RI α on their surface. BDCA1+ DCs also show moderate expression of Fc ϵ RI α . In agreement with current literature, we could not detect significant amounts of Fc ϵ RI α on the cell surface of primary blood monocytes⁴. Interestingly, we find that BDCA1+ DCs but not basophils co-express CD23 on their cell surface. Monocytes only display low amounts of cell surface CD23. Also, the IgG-receptor profile on the three cell populations was quite different with DCs and monocytes but not basophils expressing the high-affinity IgG receptor CD64. All expressed CD32 on their surface, but only monocytes displayed detectable amounts of CD16.*

*To assess the inhibitory activity of anti-IgE antibodies on these different cell types we first removed endogenous IgE from the cell surface using disruptive anti-IgE DARPIn[®] bi53_79⁶. This treatment resulted in efficient stripping of the cells (**Response Fig. 1c**) and allowed us to reload them with various concentrations of recombinant JW8-IgE to find the optimal dose for subsequent inhibition experiments (**Response Fig. 1d**). Based on the results, we decided to use a concentration of 25 nM JW8-IgE on all cells. Pre-incubation of IgE with increasing concentrations of the anti-IgE antibodies ligelizumab and omalizumab resulted in a dose-dependent inhibition of IgE-binding to both basophils and BDCA1+ DCs (**Response Fig. 1e**). The half-maximal inhibitory concentration (IC₅₀) on basophils was significantly lower for ligelizumab compared to omalizumab, whereas no difference in IC₅₀ was observed on BDCA1+ DCs. This is most likely due to simultaneous expression of Fc ϵ RI α and CD23 on DCs. Surprisingly, IgE:anti-IgE complex formation led to significantly increased IgE binding signal on monocytes. We assume that this binding occurred through the engagement of anti-IgE:IgE immune complexes with IgG-receptors since monomeric IgE alone shows almost no binding to monocytes (**Response Fig. 1d**).*

*Taken together, our new data provide clear evidence that both ligelizumab and omalizumab block binding of IgE to cells expressing FcεR1α. Consistent with what we have reported in our original manuscript, ligelizumab is significantly more potent in inhibiting IgE-binding to FcεR1α expressing cells such as basophils (**Response Fig. 1f**). This functional advantage over omalizumab is lost, as soon as the cell co-expresses CD23 as it is the case for BDCA1⁺ DCs (**Response Fig. 1g**), although ligelizumab retains similar potency to omalizumab in these cells.*

*In the revised manuscript, we have included the new data on receptor expression and IgE inhibition in BDCA1⁺ DCs (**new Supplementary Fig. 3a-c**). We also revised the text accordingly (**pg.8, ln.164-181**). As we did not observe significant amounts of FcεR1α on CD14⁺ primary human blood monocytes, we decided leave these results out.*

R1-Comment-3: FcεRI internalization essays should be performed to analyze what the effect is of ligelizumab on FcεRI-IgE recycling in live cells (basophils/monocytes) cultured in time.

*IgE-mediated antigen uptake is a well-defined function of cells that are expressing IgE-receptors^{7,8}. In response to this reviewer’s comment we performed a time course experiment for IgE:antigen complex internalization with isolated primary basophils, BDCA1⁺ DCs and monocytes. To do so, we used IgE labeled with a pH-sensitive dye (pH-rhodamine) for which cellular uptake of the complex results in a fluorescent signal. Surprisingly, basophils rapidly and efficiently internalize the IgE:antigen complex (**Response Fig. 2a**). After 4 hours of incubation they reached a plateau in which ~65% of all basophils are positive for intracellular IgE. Also BDCA1⁺ DCs and monocytes take up the IgE:allergen complex - however, less rapidly and to lesser extent (~40% of cells after 16 hours). Based on these results we investigated whether IgE:antigen uptake can be blocked by F(ab')₂ fragments or full-length antibodies of ligelizumab and omalizumab. At 0.2-fold molarity both F(ab')₂ and full-length antibodies of ligelizumab and omalizumab inhibit IgE:allergen complex uptake in basophils (**Response Fig. 2b**). In line with previous findings, ligelizumab-mediated inhibition of IgE-binding to FcεR1α on these cells was more efficient as compared to omalizumab. Also for BDCA1⁺ DCs internalization can be blocked but only with F(ab')₂ of ligelizumab and omalizumab. Full-length anti-IgE antibodies were not inhibitory with BDCA1⁺ DCs or even increased internalization in monocytes suggesting that in DCs and monocytes inhibition of uptake via IgE-receptors is bypassed by uptake of IgG:IgE:antigen complexes by IgG-receptors. For monocytes, similar but less pronounced inhibition trends as for DCs were observed.*

*Together, we show that IgE-mediated antigen uptake is inhibited by the anti-IgE antibodies ligelizumab and omalizumab. However, cells expressing the high affinity IgG receptor FcγRI/CD64 (**Response Fig. 1b**) will still take up ligelizumab or omalizumab bound complexes via this alternative pathway.*

*In the revised manuscript, we have included the new data on IgE:antigen uptake in basophils and BDCA1⁺ DCs (**new Supplementary Fig. 3d-e**). We also revised the text accordingly (**pg.8, ln.164-181**).*

Fig. 2. Inhibition of IgE:antigen complex uptake by basophils, BDCA1⁺ DCs and monocytes. (a) Time-dependent flow cytometric quantification of IgE:antigen internalization. (b) Inhibition of IgE:antigen uptake into basophils, DCs and monocytes in the presence of an 0.2-fold molarity of ligelizumab and omalizumab IgG over a duration of 8 hours. Each experiment was performed at least twice. Data are displayed as mean \pm SEM ($n=2$ donors).

R1-Comment-4: What is the underlying mechanism of ligelizumab-induced suppression of IgE synthesis (Figure 4)? It is speculated in the discussion that antibody-dependent cellular cytotoxicity (ADCC) of B-cells may be involved, but also a direct effect via CD23 stabilization or cross-linking is possible. They should perform experiments to distinguish between these different possibilities.

We agree with the reviewer that a detailed understanding of the mechanisms underlying ligelizumab-dependent suppression of IgE production in B-cells would be of interest. However, such an investigation is quite extensive and complex and would go beyond the scope of this study. Nevertheless, we performed several experiments in response to the reviewer's comment, which helped us to exclude certain possibilities and to gather evidence for potential mechanisms.

[Redacted]

[Redacted]

*To further confirm ligelizumab-mediated suppression of IgE production of B-cells in another experimental system we cultured human PBMCs in the presence or absence of ligelizumab and omalizumab in medium containing CD40L and IL-4 (induction of IgE switch) for 12 days and subsequently quantified secreted IgE in the culture supernatants. Since previous studies have reported the involvement of FcγRIIb/CD32b on B-cells in the regulation of antibody production⁹⁻¹¹, we additionally included F(ab')₂ fragments of ligelizumab and omalizumab lacking an Fc-portion and blocked FcγRIIb in these experiments. In line with our original findings, we observed more pronounced suppression of IgE production by ligelizumab IgG as compared to omalizumab IgG (**Response Fig. 4a**). Interestingly, F(ab')₂ fragments of ligelizumab showed the same inhibition, while F(ab')₂ fragments of omalizumab had no effect on IgE production. These results indicate that ligelizumab-induced suppression of IgE production is independent of Fc-mediated effector mechanisms. When we additionally blocked FcγRIIb using 500 nM of an anti-FcγRIIb DARPin® protein¹² (**Response Fig. 4b**) omalizumab entirely lost its inhibitory activity, while ligelizumab-mediated suppression of IgE production remained unaltered. This provides further evidence that ligelizumab inhibits B-cells independently of FcγRIIb engagement.*

Fig. 4. Assessment of FcγRIIb-involvement in suppression of IgE production by anti-IgE antibodies. (a) Purified human PBMCs were incubated for 12 days with CD40L and IL4 in the presence or absence of ligelizumab and omalizumab IgG or F(ab')₂ fragment. As a positive control rituximab IgG has been included. (b) FcγRIIb was blocked using an anti-CD32b molecule. Each experiment was performed at least twice. Data are displayed as mean ± SEM (n=6). Statistical comparison was calculated by one-way ANOVA with multiple comparisons against the untreated control. *P < 0.05, ***P < 0.005, **P < 0.01, ns = not significant.

Given these new insights into potential mechanisms involved in the ligelizumab-mediated suppression of IgE production in B-cells, we have revised both text and figures in the manuscript. We removed our hypothesis on ADCC induction by ligelizumab via engagement of CD23:IgE complexes and included evidence for an FcγRIIb-independent mechanism (new revised Fig. 5) that involves CD23 clustering on the B-cell surface (new revised Fig.4), which has previously been described as mechanism for suppression of IgE production by others^{13,14}.

R1-Comment-5: In the mouse model, there is no human CD23 present. If this is the reason why Ligelizumab is more effective than Omalizumab in vivo, then the question is how relevant this finding is. Have the authors performed other in vivo studies (other allergic (humanized) mouse models) to exclude this?

The reviewer raises an interesting point. We would like to highlight that the passive systemic anaphylaxis (PSA) mouse model performed in this study is fully dependent on and mediated by FcεR1α on allergic effector cells but independent of CD23^{15,16}. The rationale of using this model was to verify the efficacy of the two anti-IgE antibodies to inhibit the IgE:FcεR1α interaction in an in vivo setting and confirm the in vitro data that ligelizumab is more potent in inhibiting IgE-binding to FcεR1α. CD23 is thought to be important in the sensitization phase of an allergy but not in the effector phase of an anaphylactic response. Therefore, the information one would gain from investigating the possible contribution of CD23-dependent mechanisms in a passive anaphylaxis model would be very limited. Additionally, human CD23 transgenic mice have not been described so far and the generation of a triple transgenic mouse expressing human IgE, the human FcεR1α as well as human CD23 would clearly go beyond the scope of this study. Recently, several humanized mouse models have been developed based on immunodeficient mice that were repopulated with either human PBMCs or CD34⁺ hematopoietic stem cells. However, it has been noted that B-cells in these mice show intrinsic defects in antibody class switching¹⁷. Furthermore, a study assessing asthmatic airway inflammation in humanized mice revealed that this model is of limited use for the investigation of IgE-mediated allergic reactions, since the mice showed extremely low to non-existent levels of IgE¹⁸. Even in a study assessing active sensitization to peanut allergy using a humanized mouse model the authors found surprisingly low levels of allergen-specific IgE (~1 IU/L corresponding to 2.4 ng/ml), which would make it tremendously difficult to qualitatively and quantitatively compare different anti-IgE antibodies in this system¹⁹. Therefore, we strongly feel that there is currently no better or more suitable model available for the comparison of omalizumab and ligelizumab in an in vivo mouse model.

We would like to point out to the reviewer, that the in vivo efficacy of ligelizumab and omalizumab have recently been compared in a phase III clinical trial with human chronic spontaneous urticaria patients (CSU). The results of this study were published in the New England Journal of Medicine just a few days ago²⁰. They confirm our observation that ligelizumab is more potent in suppressing disease activity in these patients compared to omalizumab. CSU is a condition that is crucially dependent on the IgE-FcεRIα signaling axis in mast cells and basophils.

Other points:

R1-Comment-6: The reported affinity (Kd) of omalizumab in Table 1 is 10x times higher than previously reported work [E. Cohen mAbs 2014]. Can the authors comment on this?

*In their work, Cohen et al.²¹ performed interaction measurements based on the KinExA technology, whereas we measured binding kinetics using surface plasmon resonance technology (SPR). In their KinExA experiment Cohen et al. pre-incubated IgE with omalizumab in solution and then applied the equilibrated binding partners to a bead-column carrying immobilized IgE-Fc₃₋₄ fragments. From our own SPR measurements provided in the original submission we know, that the affinity value of omalizumab to IgE-Fc₃₋₄ fragments is roughly 10-times higher than to full length IgE (**original Fig. 1b and e and Table 1 and 2**). Thus, the experimental setups and used technologies explains the differences in reported Kd.*

R1-Comment-7: For most experiments, it is not clearly stated how many independent experiments were performed. Please include this information in the legend, including supplementary legends.

As requested by the reviewer, we have included the number of technical or biological replicates or experiment repetitions in the figure legends of the revised manuscript where appropriate.

R1-Comment-8: It seems that the in vivo experiment (Fig. 4) has been performed once with only 3 mice/group??

*In response to the reviewer's comment we have repeated this in vivo experiment and now show pooled data of two independent experiments with n=6 mice per group (**revised Fig. 4d-f**). We also indicate this in the revised figure legend. The results and conclusions remain the same.*

Reviewer #2 (Immune structure, FcR)(Remarks to the Author):

IgE antibodies and its high affinity receptor FcεRI are the fundamental components of allergic response, and their interaction leads to immediate hypersensitivity reactions against various allergens. In addition to high affinity receptor FcεRI, IgE can also bind to another receptor on mast cells, namely CD23. Unlike the high affinity receptor FcεRI that is mainly responsible for IgE-mediated mediator release, CD23 might have both positive and negative regulation effects on IgE production. The investigation of the interaction between IgE and its receptors has been a fundamental question in structural immunology. In the meanwhile, the advance in the understanding of these

structural mechanisms also supports the therapeutic strategy to block the IgE-IgE receptor interaction to alleviate allergic response.

For example, therapeutic antibody, Omalizumab, significantly decreases serum IgE in patients and has been approved for clinical treatments for allergic asthma. However, this type of blocking antibodies prevents IgE from its binding of both FcεRI and CD23 and thus undermines the potential beneficial effect of CD23. Therefore, there is a unique request to raise antibodies to differentially block IgE binding to its high affinity and low affinity receptors. Toward this goal, a new high affinity anti-IgE monoclonal antibody (ligelizumab; QGE031) showed greater efficacy than Omalizumab on inhaled and skin allergen responses in patients with mild allergic asthma, representing a promising next generation therapeutic antibody to suppress FcεRI-mediated allergic response.

In the current manuscript, the authors characterized the binding of ligelizumab to IgE and further determined its binding epitope on IgE using X-ray crystallography. Despite the intermediate resolution, their results revealed that ligelizumab binds to free IgE at sites with a major overlapping with FcεRI and a minor overlapping with CD23. However, ligelizumab was not able to displace cell-surface bound IgE like Omalizumab. Instead, it could bind to IgE:CD23 on the cell surface, suggesting a unique process to mediate its therapeutic effects. Although the characterization the binding of ligelizumab to IgE in the presence or absence of FcεRI was well carried out, the binding of ligelizumab to IgE:CD23 complexes in solution or on the cell surface was not complete and the physiological effect of such binding needs to be further investigated to establish a solid foundation for understanding the highly efficient suppression of FcεRI-mediated allergies. Therefore, a revision is necessary before the current manuscript could be published in the journal of Nature Communication to shed new lights in the field to further develop this class of antibodies toward better therapeutic effects.

R2-Comment-1: In Fig2, the authors showed that both Ligelizumab can block the binding of IgE to both FcεRI and CD23, which agrees with their structure observation that Ligelizumab's binding epitope on IgE is overlapping with those of FcεRI and CD23, suggesting mutually exclusive bindings. Furthermore, the BIAcore studies in Fig 3 confirmed that Ligelizumab can't bind to and dissociate IgE bound onto FcεRI. A similar BIAcore study should be carried out to demonstrate the binding among IgE, CD23, and Ligelizumab to support the idea that Ligelizumab could bind to IgE:CD23 complexes.

*As suggested by the reviewer, we performed additional BIAcore as well as ELISA experiments to demonstrate that ligelizumab can interact with CD23-bound IgE. For the BIAcore assay we immobilized 3000 response units (RU) recombinant human CD23 on a CM5 chip surface. Injection of different IgE concentrations (31.25-500 nM) resulted in a dose-dependent binding (**Response Fig. 5a**). The fitting of the measured binding curves with a two-state reaction model was nearly perfect. To test binding of ligelizumab and omalizumab to CD23:IgE complexes (**Response Fig. 5b**), we added IgE at 125 nM and subsequently injected different concentrations of the anti-IgE antibodies (31.25-500 nM). In line with our results using RPMI8866 cells (**Response Fig. 3**) both anti-IgE antibodies induced dissociation of IgE from CD23 as judged by the rapidly decreasing response units compared to buffer control. Since the drop in*

RU signal represents a delta of simultaneous anti-IgE binding and IgE removal we used an anti-IgG antibody to specifically detect CD23:IgE:anti-IgE trimeric complexes. In agreement with the data on RPMI8866 cells (**Response Fig. 3c**) we detected significant binding of ligelizumab to CD23:IgE complexes (**Response Fig. 5c**). Additionally, we performed an ELISA to confirm these results. A concentration of 50 nM IgE showed steady binding to 300 nM immobilized CD23 (**Response Fig. 5d**). Addition of 10 nM ligelizumab to this CD23:IgE complex revealed significant amount of binding when detected by anti-IgG-HRP staining (**Response Fig. 5e**) but only minor binding with omalizumab similar to the data with CD23+ cells.

Together all three experimental systems (RPMI8866 cells, BIAcore and ELISA) indicate that ligelizumab recognizes CD23:IgE complexes, whereas there is only minor binding of omalizumab detectable. We included the BIAcore data together with the cellular binding assays in the revised manuscript (**new revised Fig. 4**) and edited the text accordingly (**pg.13, ln.267-275**).

Fig. 5. Assessment of ligelizumab binding to CD23:IgE complexes by BIAcore and ELISA. (a) Sensorgrams of IgE titration on immobilized recombinant human CD23 using BIAcore. (b) Sensorgrams of ligelizumab and omalizumab IgG titration on CD23:IgE complexes. (c) Detection of remaining ligelizumab and omalizumab binding to CD23:IgE complexes. (d) IgE binding of IgE on immobilized human recombinant CD23 by ELISA ($n=3$) (e) Detection of ligelizumab and omalizumab binding to CD23:IgE complexes ($n=3$). Data are displayed as mean \pm SEM. Statistical comparison was performed by Student's t-test. ****P < 0.001, ***P < 0.005.

R2-Comment-2: In Fig4e and Fig4f, the MFI of cell surface IgE in the presence of Ligelizumab was ~ 5000 and 750, respectively. If the data were obtained in the same experiment, it suggests that a very small portion of cells could be stained with streptavidin (for IgE) and anti-human Ig light chain (for Ligelizumab). Therefore, the authors' conclusion that Ligelizumab was able to bind to IgE:CD23 complex is shaky. The detailed FACS staining dot profile should be provided. In addition, a microscopic investigation should be carried out to provide further information to support authors' proposal.

We hope that our data in **Response Fig. 5** could convince the reviewer that *ligelizumab* has the ability to bind *CD23:IgE* complexes, although – and we fully agree with the reviewer – that this binding is limited due to partial displacement of *IgE:CD23* complexes and inhibition of *IgE* binding to *CD23*. Here we additionally provide flow cytometry data dot plot graphs and the frequency of *RPMI8866* cells that displayed *ligelizumab* on their surface as requested by the reviewer (**Response Fig. 6a**).

Fig. 6. Assessment of ligelizumab binding to CD23:IgE complexes on RPMI8866 cells. (a) Cell surface IgE levels on RPMI8866 leukemia B-cells sensitized with JW8-IgE and incubated with ligelizumab, omalizumab or control IgG for 30 minutes with prior blocking of CD23 or not. **(b)** Binding of ligelizumab, omalizumab or control IgG to JW8-IgE sensitized RPMI8866 leukemia B-cells with or without prior blocking of CD23. Background signals were subtracted from measured values. **(c)** Percentage RPMI8866 cells displaying CD23:IgE complex bound ligelizumab or omalizumab IgG. **(d)** Representative flow cytometry dot plots showing percentage of IgG⁺ RPMI8866 cells. **(e)** Image stream analysis of surface CD23, IgE and IgG on RPMI8866 cells sensitized with IgE alone or subsequently treated with ligelizumab, omalizumab or isotype control IgG. Each experiment was performed at least twice. Data are displayed as mean ± SEM (n=2).

Furthermore, we show microscopic evidence from image stream measurements that ligelizumab is bound to the surface of RPMI8866 cells that were sensitized with IgE (Response Fig. 6b) and moreover revealing cluster staining indicating micro-aggregation of CD23:IgE complexes by ligelizumab on the surface of cells.

We included this new data in the revised manuscript (new revised Fig. 4) and edited the text accordingly (pg.14, ln.285-293).

Reviewer #3 (Antibody response, immune structure)(Remarks to the Author):

The authors have used crystallography, SPR, ELISA, various cell-based in vitro assays, and a mouse model to compare the activities and modes of action of two anti-IgE antibodies of interest for suppression of allergic responses in humans, omalizumab and ligelizumab. The authors distinguish allergy signalling through two cell surface receptors, Fc_εRI and CD23. Their data demonstrate that ligelizumab has higher affinity and avidity for IgE than omalizumab and prefers the open conformation of IgE C-ε3, whereas omalizumab binds similarly to both open and closed conformations. Their crystal structure of ligelizumab ScFv bound to an IgE Fc3-4 fragment reveals the binding mode, shows how it differs from that of omalizumab, and is consistent with ligelizumab favoring a more open conformation. ELISA and flow cytometry showed that ligelizumab blocked IgE binding to Fc-εRI more efficiently than omalizumab, but in contrast, ligelizumab was inferior to omalizumab for blocking IgE binding to CD23. The authors postulated structural mechanisms for this difference in specificity, based on analysis of their crystal structure and others in the literature. The authors showed that the two antibodies act differently in other potentially important ways too: (i) omalizumab but not ligelizumab can accelerate dissociation of IgE from the surface of FC-εRI-expressing cells; (ii) while omalizumab binds to preformed complexes of FC-εRI with IgE Fc3-4, ligelizumab disrupts such complexes; (iii) ligelizumab but not omalizumab removes cell surface IgE-Fc3-4 in a dose-dependent manner; (iv) omalizumab but not ligelizumab activates basophils re-sensitized with IgE-Fc3-4 in a dose-dependent manner. Finally, in terms of functional efficacy, the authors showed that ligelizumab has higher potency in inhibiting basophil activation; provided B cell ELISpot evidence that ligelizumab is superior to omalizumab for suppression of IgE production by B cells; and showed that ligelizumab is superior for protection against anaphylaxis in a mouse model.

The work addresses questions of significant relevance to treatment of human allergic diseases. The manuscript is well-written, the conclusions are carefully drawn and seem well supported by the data. The work provides a solid pre-clinical foundation on which to assess clinical studies of ligelizumab, and potentially opens the door to future therapies that further optimize the parameters explored here. Areas of further exploration would certainly include the distinctions noted in interfering with Fc_εRI vs CD23 signalling and the potential that omalizumab cross-linking of complexes of Fc-εRI and IgE-Fc3-4 may be related to the observed low level of anaphylaxis of omalizumab-treated patients, whereas ligelizumab may avoid that complication by not binding to such complexes if they do in fact exist in vivo.

Minor

R3-Comment-1: page 3. binding of IgE to FCepsilonRI is stated to have high affinity of KD 0.1 nM. Is this a monovalent KD? Please clarify if bivalent avidity is in play here.

The affinity of IgE to FcεR1α refers to a monovalent interaction that has been previously reported²². We would like to point out to the reviewer, that the binding between IgE and its high-affinity receptor FcεR1α occurs in an asymmetric way²³. IgE contacts the receptor at two independent binding sites adopting a conformation resulting in a 1:1 stoichiometry. Bivalent binding events between IgE and FcεR1α are structurally not possible.

R3-Comment-2: page 3. "...cross-linking of FCεRI-bound IgE..." should that not be "...cross-linking of IgE-bound FCεRI..."?

We feel that these terms can be used interchangeably, since both IgE and the receptor get cross-linked upon binding of the allergen. Nevertheless, we edited this sentence, as suggested by the reviewer (pg. 3, ln. 46).

R3-Comment-3: page 5. A KD of 18 pM is given for the interaction of ligelizumab with IgE. The format of the SPR measurement suggests that bivalent avidity is in play here. Please clarify.

We agree with the reviewer that bivalent interactions in this experimental setup seem to occur to some degree with intact antibody, even though the density of immobilized IgE on the chip surface was intentionally chosen to be very low (100 RU). The fact that the affinity of the Fab fragments of ligelizumab is about two times lower than for the F(ab')₂ and full-length antibody hints towards a certain bivalent interaction. Thus, comparing the affinities of both Fab fragments seems more appropriate. We revised the Figure (new Fig. 1a and b) and text (pg. 5, ln. 91-94) accordingly.

R3-Comment-4: page 5. The affinity of the ligelizumab Fab fragment for IgE is given as 35 nM, which suggests to this reviewer that the 18 pM KD given for the full ligelizumab IgG indeed includes the effect of bivalent avidity. Yet, the authors say "The affinity of the ligelizumab Fab fragment was 35 nM, indicating that intact antibody binding was not avidity driven." This last statement seems incorrect. Please clarify.

Due to the rather small difference between Kd for Fab and F(ab')₂ we meant to communicate that the affinity of ligelizumab is not mainly avidity driven. However, as as stated above, we have edited the text and thank the reviewer for this helpful expert feedback.

R3-Comment-5: page 9. when comparing the binding modes of ligelizumab/IgE-Fc and CD23/IgE-Fc, the authors distinguish between "steric overlap" and "competition for IgE surface subsites". But these are not mutually exclusive concepts -- in order for CD23 and ligelizumab to bind to the same subsites on IgE, some steric overlap would obviously occur.

The reviewer raises a good point and we have revised the manuscript in this section to clarify this presentation of the results. Although ligelizumab shares some residue contacts on IgE with CD23, the amount of steric overlap is significantly lower as compared to omalizumab. We have made this clearer for both FcεRI and CD23 inhibition in the revised text and included a supplementary Figure (**Supplementary Fig. 4**) to emphasize these points.

References:

1. Lindberg, R. E. & Arroyave, C. Levels of IgE in serum from normal children and allergic children as measured by an enzyme immunoassay. *Journal of allergy and clinical immunology* **78**, 614–618 (1986).
2. Hamilton, R. G. & Oppenheimer, J. Serological IgE Analyses in the Diagnostic Algorithm for Allergic Disease. *J Allergy Clin Immunol Pract* **3**, 833–40– quiz 841–2 (2015).
3. Arm, J. P. *et al.* Pharmacokinetics, pharmacodynamics and safety of QGE031 (ligelizumab), a novel high-affinity anti-IgE antibody, in atopic subjects. *Clin Exp Allergy* **44**, 1371–1385 (2014).
4. Shin, J.-S. & Greer, A. M. The role of FcεRI expressed in dendritic cells and monocytes. *Cell. Mol. Life Sci.* **72**, 2349–2360 (2015).
5. Greer, A. M. *et al.* Serum IgE clearance is facilitated by human FcεRI internalization. *J Clin Invest* **124**, 1187–1198 (2014).
6. Eggel, A. *et al.* Accelerated dissociation of IgE-FcεRI complexes by disruptive inhibitors actively desensitizes allergic effector cells. *J Allergy Clin Immunol* **133**, 1709–19.e8 (2014).
7. Sokol, C. L., Barton, G. M., Farr, A. G. & Medzhitov, R. A mechanism for the initiation of allergen-induced T helper type 2 responses. *Nat Immunol* **9**, 310–318 (2008).
8. Mudde, G. C., Hansel, T. T., Reijssen, F. C. V., Osterhoff, B. F. & Bruijnzeel-Koomen, C. A. F. M. IgE: an immunoglobulin specialized in antigen capture? *Immunol Today* **11**, 440–443 (1990).
9. Horejs-Hoeck, J., Hren, A., Mudde, G. C. & Woisetschläger, M. Inhibition of immunoglobulin E synthesis through Fc gammaRII (CD32) by a mechanism independent of B-cell receptor co-cross-linking. *Immunology* **115**, 407–415 (2005).
10. Daëron, M. *et al.* The same tyrosine-based inhibition motif, in the intracytoplasmic domain of Fc gamma RIIB, regulates negatively BCR-, TCR-, and FcR-dependent cell activation. *Immunity* **3**, 635–646 (1995).
11. Chu, S. Y. *et al.* Reduction of total IgE by targeted coengagement of IgE B-cell receptor and FcγRIIb with Fc-engineered antibody. *J Allergy Clin Immunol* **129**, 1102–1115 (2012).
12. Zellweger, F. *et al.* A novel bispecific DARPin targeting FcγRIIb and FcεRI-bound IgE inhibits allergic responses. *Allergy* (2016). doi:10.1111/all.13109
13. Liu, C. *et al.* CD23 can negatively regulate B-cell receptor signaling. *Sci Rep* **6**, 25629 (2016).
14. Luo, H. Y., Hofstetter, H., Banckereau, J. & Delespesse, G. Cross-linking of CD23 antigen by its natural ligand (IgE) or by anti-CD23 antibody prevents B lymphocyte proliferation and differentiation. *J Immunol* **146**, 2122–2129 (1991).
15. Dombrowicz, D. *et al.* Anaphylaxis mediated through a humanized high affinity IgE receptor. *J Immunol* **157**, 1645–1651 (1996).
16. Dombrowicz, D., Flamand, V., Brigman, K. K., Koller, B. H. & Kinet, J. P. Abolition of anaphylaxis by targeted disruption of the high affinity immunoglobulin E receptor alpha chain gene. *Cell* **75**, 969–976 (1993).
17. Ito, R., Takahashi, T., Katano, I. & Ito, M. Current advances in humanized mouse models. *Cell. Mol. Immunol.* **9**, 208–214 (2012).
18. Ito, R. *et al.* A humanized mouse model to study asthmatic airway inflammation via the human IL-33/IL-13 axis. *JCI Insight* **3**, (2018).
19. Burton, O. T. *et al.* A humanized mouse model of anaphylactic peanut allergy. *J Allergy Clin Immunol* **139**, 314–322.e9 (2017).
20. Maurer, M. *et al.* Ligelizumab for Chronic Spontaneous Urticaria. *N Engl J Med* **381**, 1321–1332 (2019).
21. Cohen, E. S. *et al.* A novel IgE-neutralizing antibody for the treatment of severe uncontrolled asthma. *MAbs* **6**, 756–764 (2014).

22. Pennington, L. F. *et al.* Structural basis of omalizumab therapy and omalizumab-mediated IgE exchange. *Nat Commun* **7**, 11610–12 (2016).
23. Garman, S. C., Kinet, J. P. & Jardetzky, T. S. Crystal structure of the human high-affinity IgE receptor. *Cell* **95**, 951–961 (1998).

REVIEWERS' COMMENTS:

Reviewer #1 (Remarks to the Author):

The authors have submitted an excellent revision. All my comments have been addressed and I recommend acceptance of the manuscript.

Reviewer #2 (Remarks to the Author):

In the revision, the authors tried to address the concerns on the binding of Ligelizumab to CD23-IgE complex by BIAcore, ELISA, cell surface binding and cellular stimulation, which was also raised by other reviewers. The addition of data strengthened the manuscript significantly.

The BIAcore binding data showed that the binding of Ligelizumab to CD23-IgE complex could not stabilize the pre-formed CD23-IgE complex judging by the accelerated dissociation. This is unexpected as RU usually increases if a tertiary complex formed upon the antibody binding. It is intriguing to see whether pre-formed Ligelizumab-IgE complex could bind to CD23 if the steric overlap of their binding sites is lower enough to allow tertiary binding.

In addition, the FACS profile is somewhat ambiguous since the whole profile shifted and only 35% but not the whole population displayed Ligelizumab:CD23:IgE triple complex. A simultaneous multiple color FACS analysis should be better to characterize such complex binding.

The reviewers understand that to precisely determine the exact mechanism by which Ligelizumab offers better efficacy is beyond the scope of current manuscript.

Given the remaining caveat in the revised manuscript, the author should state the other possibilities that are also consistent with their data.

Reviewer #3 (Remarks to the Author):

The authors have satisfactorily addressed my comments.

We would like to thank the editor and the three referees again for re-evaluating the revised version of our manuscript carefully and agree that the provided expert opinion of all three reviewers has helped to further improve the quality of this study. Please find below a point-by-point reply below (in blue) to the remaining concerns.

Reviewer #1 (Remarks to the Author):

The authors have submitted an excellent revision. All my comments have been addressed and I recommend acceptance of the manuscript.

Reviewer #2 (Remarks to the Author):

In the revision, the authors tried to address the concerns on the binding of Ligelizumab to CD23-IgE complex by BIAcore, ELISA, cell surface binding and cellular stimulation, which was also raised by other reviewers. The addition of data strengthened the manuscript significantly.

We are happy to hear that the reviewer likes the additional data.

The BIAcore binding data showed that the binding of Ligelizumab to CD23-IgE complex could not stabilize the pre-formed CD23-IgE complex judging by the accelerated dissociation. This is unexpected as RU usually increases if a tertiary complex formed upon the antibody binding. It is intriguing to see whether pre-formed Ligelizumab-IgE complex could bind to CD23 if the steric overlap of their binding sites is lower enough to allow tertiary binding.

Fig. 1 Inhibition of IgE-binding to CD23. Recombinant human JW8-IgE was incubated with different concentrations of ligelizumab (a) or omalizumab IgG (b). IgE binding responses to immobilized recombinant human CD23 on the chip surface were measured by SPR. Each color in the sensorgram refers to an individual measurement cycle for the indicated concentration of anti-IgE antibodies. (c) Comparison between ligelizumab and omalizumab IgG mediated inhibition of IgE binding to CD23. Data refers to the binding signal at time point 180 seconds from the SPR measurements.

As the reviewer states correctly, we report that ligelizumab accelerates the dissociation of pre-formed CD23:IgE complexes. However, in **original Fig. 4** we provide multiple lines of evidence that ligelizumab engages CD23-bound IgE and forms tertiary complexes. Our data strongly suggest, that even though some receptor ligand disruption occurs upon ligelizumab treatment there is still enough CD23-bound IgE remaining that will be recognized by ligelizumab. From our structural analysis, we conclude that the binding sites of ligelizumab and CD23 on IgE show only minor steric overlap (**Supplementary Fig. 4**). Nevertheless, ligelizumab inhibits IgE-binding to CD23 dose-dependently when pre-complexed with IgE in ELISA and on cells (**original**

Fig. 2d and f). Upon reviewer’s request, we also provide BIAcore data supporting the finding that both anti-IgE antibodies inhibit IgE-binding to CD23, with omalizumab showing higher efficacy (Response Fig. 1). As stated in the manuscript, we suggest that this is due to the steric constraint that ligelizumab binding puts on IgE. The structural data indicates that in the ligelizumab complex, the IgE Cε3 dimer adopts an open conformation and is therefore incompatible with CD23 binding - which requires a closed Cε3 conformation.

In addition, the FACS profile is somewhat ambiguous since the whole profile shifted and only 35% but not the whole population displayed Ligelizumab:CD23:IgE triple complex. A simultaneous multiple color FACS analysis should be better to characterize such complex binding.

Fig. 2 Formation of tertiary CD23:IgE:anti-IgE complexes on RPMI8866 cells. RPMI8866 cells were either sensitized with recombinant human JW8-IgE (a), left unsensitized (b), or treated with a CD23-blocking agent before sensitization with IgE (c). Multi-color flow cytometry analysis is shown for all three conditions. Cells were gated via SSC-A and FSC-A to get rid of cell debris. Subsequently, CD19 and IgE surface levels were determined. The binding signal of ligelizumab and omalizumab IgG is shown in a histogram plot against an isotype control antibody.

In response to this comment we provide the full multi-colour staining profile of the flow cytometry experiment in original Fig. 4f for the reviewer’s reference. It shows that trimeric complexes (CD23:IgE:anti-IgE antibody) on the surface of RPMI8866 cells can only be observed upon ligelizumab but not omalizumab treatment (Response Fig. 2a). In the absence of IgE ((Response Fig. 2b and c) neither ligelizumab nor omalizumab show binding. Isotype staining with an unspecific human IgGκ1 is

*included in the histograms (red) as negative control. Shifting of the whole cell population as depicted in **original Fig. 4f** or the **Response Fig. 2** is nothing unusual, given that a homogenous cell line has been used in these experiments. First, there is a normal distribution of IgE on the cell surface. Additionally, we showed in **original Fig. 4b and d** that anti-IgE antibodies displace a significant amount of IgE from the cells. Therefore, the shift in mean fluorescent intensity might not be as prominent as one would expect. Due to space limitations, we did not include all this data in the original manuscript. Further, we would like to point out, that the image stream data in **original Fig. 4h and i**, which is also a flow cytometry based method, clearly demonstrates that ligelizumab binds and co-localizes with IgE on the surface of RPMI8866 cells.*

The reviewers understand that to precisely determine the exact mechanism by which Ligelizumab offers better efficacy is beyond the scope of current manuscript.

We thank the reviewer for his agreement. Current studies detailing molecular mechanisms are ongoing.

Given the remaining caveat in the revised manuscript, the author should state the other possibilities that are also consistent with their data.

We are not clear about this comment and assume that the reviewer is referring to the different therapeutic potentials of the two anti-IgE antibodies. In the manuscript, we state the following: "...the two anti-IgE antibodies display different abilities to inhibit IgE interactions with FcεRI and CD23 and feature a qualitatively distinct inhibition profile. Consequently, they greatly differ in their functional activities in blocking effector cell activation and IgE synthesis. While the increased affinity of ligelizumab for IgE explains superiority over omalizumab regarding neutralization of free serum IgE, we have identified an additional mode of action for ligelizumab through the inhibition of IgE production, which may provide additional therapeutic benefit. We observe that ligelizumab is more efficient in suppressing FcεRI-dependent allergic reactions in an in vivo model, while omalizumab may have advantages in blocking antigen presentation and transport processes that are dependent on IgE:CD23 interactions."

Concerning potential molecular mechanisms of ligelizumab-mediated suppression of IgE-production in B-cells, we excluded the induction of ADCC by ligelizumab and the suppression of IgE-synthesis through engagement of inhibitory receptor FcγRIIb. As pointed out in the previous response to the reviewers and in the manuscript, our data strongly suggests a B-cell intrinsic mechanism. However, as mentioned above the exact mode-of-action needs to be determined in follow-up studies.

Reviewer #3 (Remarks to the Author):

The authors have satisfactorily addressed my comments.